# The cellular coding of temperature in the mammalian cortex

M. Vestergaard[1,2,5 ✉], M. Carta[1,2,3,5 ✉], G. Güney[1,2,4] & J. F. A. Poulet[1,2 ✉]

Temperature is a fundamental sensory modality separate from touch, with dedicated receptor channels and primary afferent neurons for cool and warm[1–3]. Unlike for other modalities, however, the cortical encoding of temperature remains unknown, with very few cortical neurons reported that respond to non-painful temperature, and the presence of a 'thermal cortex' is debated[4–8]. Here, using widefield and two-photon calcium imaging in the mouse forepaw system, we identify cortical neurons that respond to cooling and/or warming with distinct spatial and temporal response properties. We observed a representation of cool, but not warm, in the primary somatosensory cortex, but cool and warm in the posterior insular cortex (pIC). The representation of thermal information in pIC is robust and somatotopically arranged, and reversible manipulations show a profound impact on thermal perception. Despite being positioned along the same one-dimensional sensory axis, the encoding of cool and that of warm are distinct, both in highly and broadly tuned neurons. Together, our results show that pIC contains the primary cortical representation of skin temperature and may help explain how the thermal system generates sensations of cool and warm.

A fundamental question in neuroscience is how the external sensory environment is represented in the cortex. In the thermal system, at present there is no consensus on how or where sensory information is encoded in the cortex. One model is that cool and warm are processed by functionally and anatomically segregated circuits, following labelled-line principles seen in primary sensory afferent neurons and spinal circuits[9–17] resulting in cool- and warm-selective cortical neurons (Fig. 1a, top). Another is that cool and warm are integrated in the thermal pathway resulting in cortical neurons with a continuous, graded representation of temperature (Fig. 1a, bottom). Testing these models requires the identification of thermally responsive neurons in the cortex.

Psychophysical studies have shown that cooling and warming elicit distinct sensations, influenced by stimulus amplitude, duration, dynamics, body part and adapted skin temperature[2,18]. Analogously to other modalities, these features should be represented in a cortical region dedicated to thermal processing. Moreover, manipulation of its activity should influence thermal perception. A number of cortical regions have been suggested to be involved in thermal processing, including the primary somatosensory cortex (S1)[6–8] and the pIC[5,19,20] (Fig. 1b). However, no study has identified a cortical area with a cellular representation of cool and warm and a reversible impact on perception.

## pIC contains a somatotopic map of cool and warm

To examine thermal processing in S1 and pIC, we carried out widefield calcium imaging through a glass window in paw-tethered, awake mice

expressing the calcium indicator GCaMP6s in cortical excitatory neurons (Fig. 1b,c and Extended Data Fig. 1). We presented an 8-kHz acoustic stimulus to locate the anterior auditory field in the auditory cortex and the small auditory insular field that border the somatosensory regions of pIC[21–23], and went on to carry out post hoc histology (Extended Data Fig. 1). Next, we delivered 2-s thermal stimuli to the forepaw glabrous skin through a Peltier element held at an adapted temperature (AT) of 32 °C. Whereas 10 °C (32 °C to 22 °C) cooling stimuli triggered reliable changes in fluorescence in S1, 10 °C warming (32 °C to 42 °C) did not (Fig. 1d). By contrast, both cooling and warming stimuli triggered robust, large-amplitude responses in pIC (Fig. 1e). Local pharmacological inactivation of either S1 or pIC during imaging abolished the thermal response in the injected region but not in the untreated region (Fig. 1f,g), indicating that pIC and S1 receive parallel streams of thermal input.

Perceptual thresholds for cool and warm covary across the body[24], suggesting that some body parts have a stronger cortical representation than others. Widefield calcium imaging of pIC showed a clear somatotopic arrangement of thermal and tactile responses anterior to the auditory cortex in all mice (Fig. 1h) with the face represented in a region anterior and ventral to the forepaw, which in turn is anterior to the hindpaw (Fig. 1h). In all mice, the forepaw had a larger-amplitude response to thermal stimuli than the hindpaw, perhaps reflecting the dominant role of the forepaw in haptic exploration (Fig. 1i). Whereas responses to cooling and touch spatially overlap in S1 (Extended Data Fig. 2; ref. [7]), they seem more separate in pIC (Fig. 1h). For all body parts tested, warming responses in pIC were delayed compared to cooling

[1]Max Delbrück Center for Molecular Medicine in the Helmholtz Association (MDC), Berlin, Germany. [2]Neuroscience Research Center, Charité-Universitätsmedizin Berlin, Berlin, Germany. [3]Univ. Bordeaux, CNRS, IINS, UMR 5297, Bordeaux, France. [4]Humboldt-Universität zu Berlin, Institut für Biologie, Berlin, Germany. [5]These authors contributed equally: M. Vestergaard, M. Carta. ✉e-mail: mikkel.vestergaard@mdc-berlin.de; mario.carta@u-bordeaux.fr; james.poulet@mdc-berlin.de

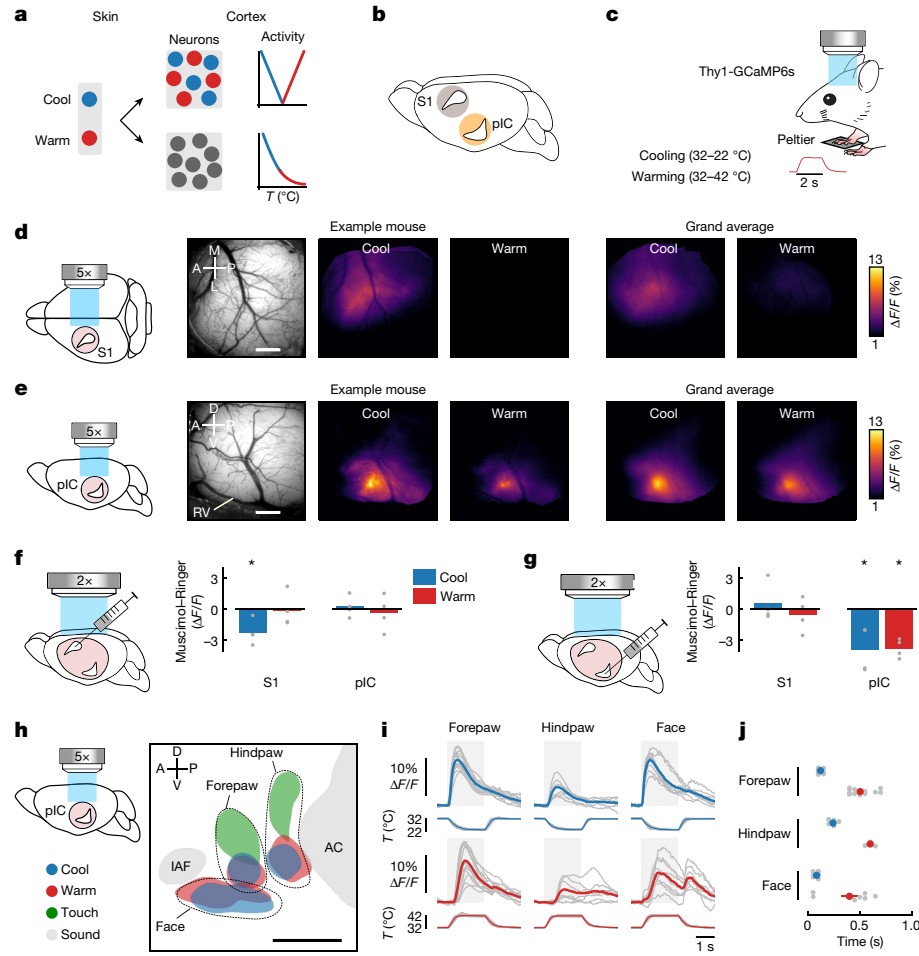

**Fig. 1 | Cortical representation of cool and warm. a**, Schematic showing segregated (top) and integrated (bottom) models of cortical thermal encoding. *T*, temperature. **b**, Mouse brain showing candidate locations of thermal cortex in primary somatosensory cortex (S1) or pIC. **c**, Schematic of an awake Thy1-GCaMP6s mouse with right forepaw on a Peltier element during widefield calcium imaging; inset shows temporal dynamics of warming stimulus. **d**, From left to right: schematic of glass window on S1 (pink circle); in vivo image of cortical surface; averaged widefield response to 10 °C cooling (32–22 °C) or warming (32–42 °C) of the forepaw in an example mouse; grand average across mice (*n* = 348 cool and 346 warm trials, 12 mice). M, medial; A, anterior; L, lateral; P, posterior. Scale bar, 500 μm. **e**, Same as **d**, but for mice with glass window implanted over pIC (*n* = 360 cool and 352 warm trials, 14 mice). RV, rhinal vein; D, dorsal; V, ventral. Scale bar, 500 μm. **f**, From left to right: schematic illustrating injection site into S1 during widefield imaging through a clear skull preparation with simultaneous imaging of S1 and pIC (pink shows field of view); bar graphs show difference in response amplitude following muscimol versus Ringer's injection. Bars indicate mean and grey filled circles indicate individual mice responses, suggesting that neural response latency is a hardwired property of the thermal system (Fig. 1j).

(*n* = 4). An asterisk indicates significant difference in S1 response following muscimol versus Ringer's injection (*P* < 0.05, two-sided paired *t*-test, see Methods for exact values). **g**, Same as **f** but showing a reduction in pIC response to thermal stimuli after pIC inactivation and no change in the S1 response (*n* = 4 mice). **h**, Somatotopic map of response locations to thermal (10 °C cool and warm), tactile (100 Hz) and acoustic (8 kHz) stimulation. Coloured area indicates peak population response averaged across mice (see Methods, *n* = 14 mice thermal forepaw, 7 thermal hindpaw, 9 thermal face, 9 touch forepaw, 6 touch hindpaw, 13 sound). Data from individual mice are aligned to peak activity of the thermal forepaw response. IAF, insular auditory field; AC, auditory cortex. Scale bar, 500 μm. **i**, Widefield responses to thermal stimulation of different body parts from the same dataset as in **h**. Grey lines show mean responses from individual mice (*n* is same as in **h**), coloured lines show population mean, and grey area indicates time from start of stimulus to end of plateau phase. **j**, Grey filled circles show response latencies from individual mice (*n* is same as in **h**, see Methods); coloured filled circles show mean ± s.e.m.

## Thermal tuning and topography of pIC neurons

Cooling and warming widefield responses in pIC overlap spatially (Fig. 1e,h). This could result from an intermingled distribution of highly tuned or broadly tuned neurons (Fig. 2a). To determine the thermal tuning of cortical neurons, we went on to carry out two-photon calcium imaging of pIC excitatory neurons in awake mice and observed robust cellular responses to cooling and warming (Fig. 2b–d). We went on to plot the normalized responses according to a thermal bias index that describes the relative response strength to cooling compared to

warming, with a value of −1 indicating a cool-only cell and +1 indicating a warm-only cell (Fig. 2d,e). The index had a U-shaped distribution, suggesting that segregated channels of tuned input drive cool and warm responses, and showed a similar probability of highly tuned neurons (cool only or warm only) and broadly tuned neurons (cool and warm responsive). In line with widefield data, most neurons in S1 were tuned to cooling only and only a tiny fraction responded to warming with delayed and inconsistent responses (Figs. 2e and 3c and Extended Data Fig. 3).

The spatial distribution of cortical neurons with respect to their functional properties is an important feature of cortical information processing[25–27]. To address whether pIC was organized thermotopically, we analysed the spatial distribution of thermally tuned neurons in pIC

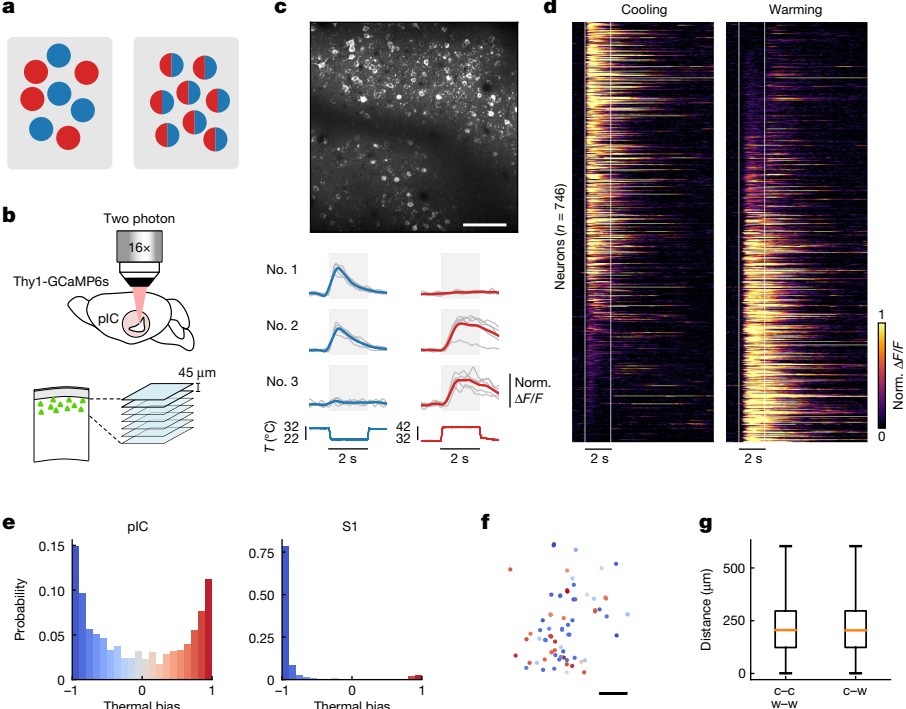

**Fig. 2 | Heterogeneous arrangement of thermally tuned neurons in the pIC.**
**a**, Schematic showing tuned (left) versus broadly tuned (right) cortical neurons.
Blue indicates cool; red indicates warm. **b**, Schematic showing two-photon
calcium imaging of pIC. Imaging started at 100 μm from the pial surface of a
Thy1-GCaMP6s mouse; 7 optical sections were acquired with intervals of
45 μm. Norm., normalized. **c**, Top, example in vivo two-photon image from
pIC. Scale bar, 100 μm. Bottom, example responses of single pIC neurons
during 10 °C cooling (blue) or 10 °C warming (red) stimuli from an AT of 32 °C
($n$ = 5 trials each). Here, and for all two-photon data figures, grey lines show
single-trial responses, coloured lines show average, and background grey
box indicates time from start of stimulus to end of plateau phase. Below,
corresponding stimulus traces. **d**, Single-cell pIC calcium responses to cooling
(left) and warming (right) stimuli. Each line represents a single neuron; responses

are normalized to the peak and sorted on the basis of the thermal bias index
with vertical white lines showing the onset and the end of the plateau phase of
thermal stimuli ($n$ = 746 neurons, 7 mice, 16 sessions). **e**, Histograms showing
the distribution of thermal bias for all responsive cells in pIC (left; $n$ is same as
in **d**) and S1 (right; $n$ = 411 neurons, 4 mice, 9 sessions). **f**, Spatial map of neurons
colour coded with their thermal bias for the representative mouse shown in **c**.
Scale bar, 100 μm. **g**, No significant difference in the distance between pairs
of neurons with similar bias (cold to cold (c–c) or warm to warm (w–w)) and
distance between pairs of neurons with opposite bias (cold to warm (c–w))
indicates no thermotopy for strongly biased neurons ($n$ is same as in **d**)
(difference of medians confidence interval (−2 μm, 5 μm), 95%, bootstrapped).
Box plots show median and interquartile range. Whiskers show minimum and
maximum values.

(Fig. 2f,g). Visual inspection showed a heterogeneous arrangement
of thermally responsive neurons (Fig. 2f). In agreement, the distance
between similarly tuned neurons (cool to cool; warm to warm) was
not significantly different to the distance between differently tuned
neurons (cool to warm; Fig. 2g). In a subset of experiments, we also used
tactile stimulation and, in line with the differing spatial arrangements
of touch and temperature, we observed a smaller fraction of thermally
responsive pIC neurons (3%) than S1 neurons (12%) that responded to
both thermal and tactile stimuli (Extended Data Fig. 3). Thus, thermal
zones of pIC contain a heterogeneous, salt–pepper-like, arrangement
of thermally responsive neurons.

## Temporal dynamics of cortical thermal responses

The skin contains discrete spots that evoke cool or warm sensations[1–3].
Notably, in vivo recordings from sensory afferent neurons, which
innervate thermal spots, have shown distinct response dynamics to
thermal stimuli, with short-latency, transient cool response mediated
by Aδ-fibres, and longer-latency, sustained firing in response to cool
and warm stimuli by C-fibres[11–14]. We went on to investigate whether
distinct temporal dynamics between cool and warm were also present in
cortical responses (Fig. 3a). In remarkable similarity to the case for the
periphery, a grand average of all significant cool and warm responding
neurons (see Methods) in pIC showed a shorter-latency (about 80 ms)

and more transient cool response compared to a longer-latency (about
320 ms) and more sustained warm response (Fig. 3b). The latency dif-
ference is present in both tuned and broadly tuned neurons (Fig. 3c),
suggesting that thermal responses are driven by similar input. Moreo-
ver, the cool response latency in S1 is similar to that in pIC, whereas the
sparse warm responses in S1 are substantially delayed and more variable
compared to pIC warm responses (Fig. 3c and Extended Data Fig. 3).

We quantified the temporal dynamics of the thermal responses in pIC
by computing a duration index that measures the change in response
level at the end of the stimulation period compared to the initial peak
value (Fig. 3d). The bimodality of the distribution of duration index
for cool responses suggests a transient and a sustained response type
(Fig. 3d). Similar results were observed for S1 cool responses (Extended
Data Fig. 4). By contrast, pIC warm responses show a broad distribution
of sustained responses (Fig. 3d). Plotting the duration index against
the response latency showed that most cool transient neurons have a
short-latency onset, whereas both cool and warm sustained neurons
show delayed onsets (Fig. 3e). In broadly tuned neurons, we observed
a similar distribution of cool and warm dynamics with cool transient
and warm sustained responses in the same neurons (Fig. 3f), together
highlighting that response dynamics are governed by sensory input
rather than intrinsic cellular properties.

Fast-onset, transient neuronal responses are thought to be reliable
indicators of stimulus change, whereas sustained responses are optimal

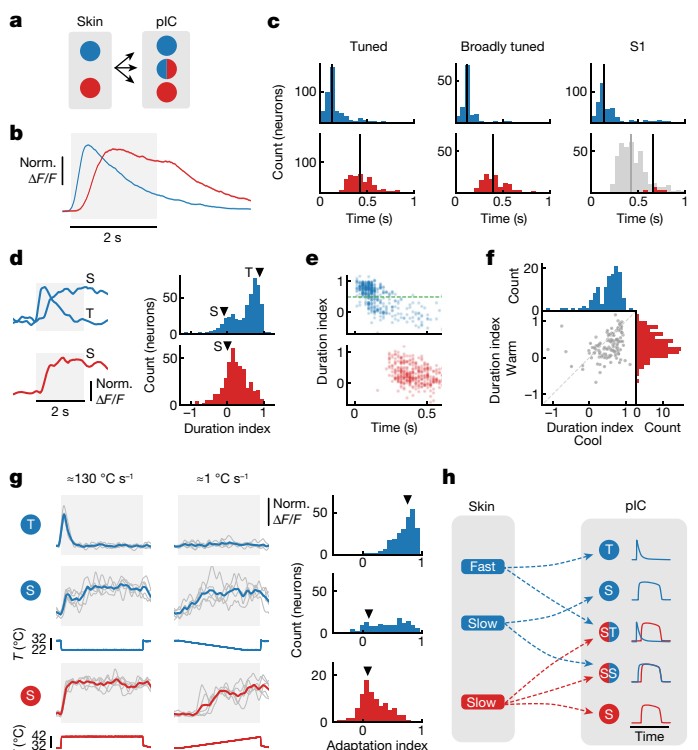

**Fig. 3 | Distinct temporal dynamics of cool and warm encoding. a**, Schematic showing transition from cool- and warm-selective skin spots to tuned and broadly tuned pIC neurons. Blue indicates cool; red indicates warm. **b**, Grand average responses (mean ± s.e.m.) to 10 °C cooling and warming responses from AT of 32 °C (n = 470 cool neurons (blue line), 401 warm neurons (red line) (see Methods); these neurons are also used in **c**–**f**). **c**, Histograms of the response latency for 10 °C cooling (top) and 10 °C warming (bottom), in (left to right) tuned neurons (n is as in **b**), broadly tuned neurons (n = 125 neurons) and S1 neurons (n = 387 cool neurons, 25 warm neurons; grey bars show comparison to tuned warm latency histogram in pIC). Vertical line represents the median. **d**, Three different example neurons (average of 5 trials) with transient (T) or sustained (S) responses to 10 °C cooling (top) or warming (bottom); histograms of duration index for cooling and warming stimuli. Arrowheads indicate the duration index of example neurons. **e**, Duration index of cool (top) and warm (bottom) responses plotted against the response latency. Green dashed line at duration index 0.5 highlights the separation between T and S neurons. **f**, Duration index of warm versus cool in broadly tuned neurons (n is as in **c** broadly tuned). **g**, Left: example traces of cool (T and S) and warm (S) neurons responding to ≈10-s stimulus at fast onset speed (left, about 130 °C s⁻¹) or at slower rate (right, about 1 °C s⁻¹) (n = 5 trials). Below, corresponding stimulus traces. Right: histograms of adaptation index for T and S neurons separated accordingly to 0.5 duration index as in **e** (n = 241 cool transient neurons, 147 cool sustained neurons, 98 warm sustained neurons, 6 mice, 11 sessions). Arrowheads indicate example neurons. **h**, Schematic model of how different channels of afferent input could drive cortical dynamics.

for absolute stimulus level encoding. We tested this hypothesis by measuring the decrease in response amplitude when a thermal stimulus is presented with different onset rates (adaption index in Fig. 3g and Extended Data Fig. 4). We observed that cool transient neurons were not activated by stimuli with a slow onset whereas sustained neurons responded similarly irrespective of the stimulus onset speed. These features were observed in highly tuned and broadly tuned pIC neurons and also in S1 cool neurons (Extended Data Fig. 4). Overall, these data

support a classic model of thermal encoding[2,3], whereby cool responses are driven by a combination of fast Aδ-fibre and slow C-fibre afferent input and warm by slower C-fibre input[11–14,18] (Fig. 3h).

## Encoding of cooling and warming amplitude

The amplitude of a sensory stimulus can be encoded in different ways: by neurons with specific preferred amplitude values; the recruiting

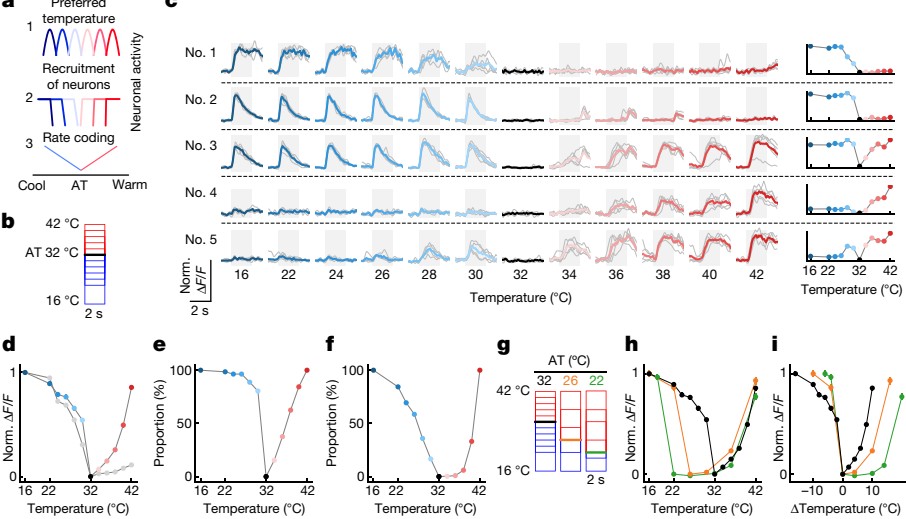

**Fig. 4 | Relative versus absolute encoding of cooling and warming.**
**a**, Schematic showing three possible cortical thermal coding schemes. Line colours indicate neuronal activity in response to cooling (blue) and warming (red). **b**, Schematic of stimulus protocol from AT of 32 °C. **c**, Left: traces of example pIC neurons responding to cooling and warming from AT 32 °C; grey lines show individual trials (5 trials per temperature), and coloured lines show mean response. Right: peak response amplitude plotted as a function of the thermal stimulus for the example neurons on the left. **d**, Summary of response amplitude plotted as a function of the thermal stimulus for entire population of

neurons (n = 746 neurons, 16 sessions, 7 mice). Coloured filled circles (pIC) and grey filled circles (S1) show mean ± s.e.m. at AT 32 °C. **e**, Same data as in **d**, but showing the proportion of recruited pIC neurons (>20% of response amplitude). **f**, Same data as in **d**, but showing the proportion of pIC neurons reaching maximum response (>80% of response amplitude). **g**, Schematic of stimulus protocols used to test impact of AT on thermal encoding. **h**, Graphs as in **d**, but for pIC neurons at all ATs studied (AT 32 °C, n is same as **d**; AT 26 °C, n = 401 neurons, 9 sessions, 5 mice; AT 22 °C, n = 448 neurons, 10 sessions, 6 mice). **i**, Same data as **h**, but plotted against change in stimulus amplitude.

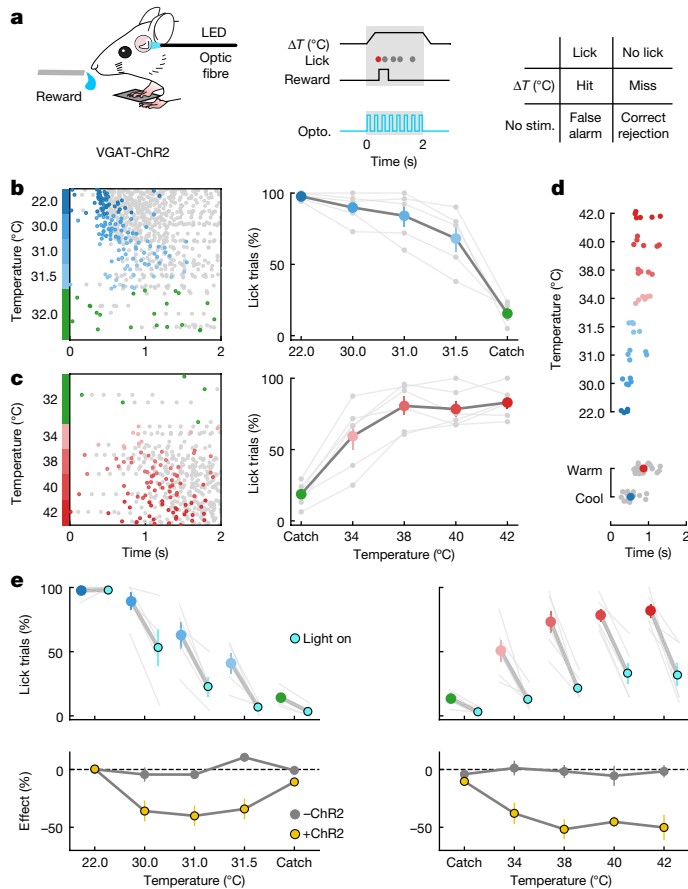

at larger amplitudes, whereas the number of activated warm neurons more gradually follows the increase of stimulus amplitude (Fig. 4e). The proportion of cool neurons reaching the maximal response amplitude is graded with stimulus amplitude, whereas warm neurons reach their maximum response amplitude only with large-amplitude stimuli (Fig. 4f). Overall, these analyses suggest that the encoding scheme for thermal amplitude is a combination of extra recruitment of neurons and graded changes of response amplitude. Among the neurons that responded significantly to warming, about 5% showed an unexpected response profile with thermometer-like properties, responding to small- but not large-amplitude cooling. These 'thermometer neurons' showed graded reporting of absolute thermal value, independent of the stimulus direction from the AT (Fig. 4c, neuron no. 5, and Extended Data Fig. 5).

The thermal responses of primary sensory afferent neurons and spinal cord neurons are influenced by the skin AT[9–17], but whether this is reflected by a change in cortical response amplitude is not known. To investigate this, we compared pIC cellular responses to stimuli of the same amplitude but from ATs of 22 and 26 °C (Fig. 4g). At lower ATs, warming stimulus amplitude had to be substantially increased to observe response amplitudes comparable to those seen at AT 32 °C, and responses were often observed only once the stimulus target temperature reached 34 °C. By contrast, reliable responses to cooling were observed for all target temperatures and ATs tested (Fig. 4h). This difference was highlighted by plotting the response amplitude against the relative stimulus amplitude, which showed a shift in the warm response curve, but not in the cool (Fig. 4i, consistent conclusions were also observed in widefield imaging data in Extended Data Fig. 6). Together these data indicate that cooling and warming amplitude is encoded in fundamentally different ways in the cortex, with warming dependent on the absolute temperature and cooling reflecting the magnitude of temperature change, consistent with findings in primary sensory afferent neurons and spinal cord neurons[9–17].

## Thermal perception is mediated by pIC

More anterior parts of insular cortex are thought to play a role in cognitive and motivational control of behaviour[28,29], whereas a central region is involved in taste perception[30]. Human lesions, microstimulation and imaging studies have suggested a role for insular cortex in thermal perception[5,20,31,32], but reversible manipulations have not been carried out and the link between thermal sensitivity and perception is unclear. To address this, we carried out optogenetic manipulations during a go/no-go thermal detection task (Fig. 5a). Thermal stimuli were delivered at random times from AT 32 °C. VGAT-ChR2 mice, with channelrhodopsin-2 (ChR2) constitutively expressed in GABAergic inhibitory interneurons, were implanted with an optical window over pIC and provided with water rewards on licking a water spout during the onset ramp or plateau phase of the stimulus (Fig. 5a and Extended Data Fig. 7). Owing to differences in sensitivity to cooling and warming in pIC (Fig. 4), we chose 10 °C, 2 °C, 1 °C and 0.5 °C cooling stimuli, whereas for warming we used 10 °C, 8 °C, 6 °C and 4 °C. Mice were able to report stimuli at all amplitudes, highlighting their acute thermal perceptual ability (Fig. 5b,c). As for the functional responses (Figs. 1 and 3), mice reported warming at longer latencies than for cooling. We went on to inhibit the activity of pIC during the stimulus phase in the same mice (Fig. 5a,e) using light pulses (12.5 mW, 20 Hz, 2 s) through a 200-µm fibre optic positioned normal to the thermal region of pIC (Fig. 5a,e). In cooling-trained mice, optogenetic inhibition of pIC suppressed the hit rates to 2 °C, 1 °C and 0.5 °C stimuli but not to 10 °C (Fig. 5e), whereas, in warming-trained mice, optogenetic inhibition suppressed the hit rates to all amplitudes (Fig. 5e). Optical stimulation efficiently inhibited neuronal responses driven by thermal stimuli in pIC (Extended Data Fig. 7). Moreover, repeating the same optical stimulation paradigm but in Thy1-GCaMP6s mice not expressing ChR2 showed no effect on thermal perception (bottom

**Fig. 5 | Thermal perception is mediated by pIC. a**, Left: schematic of thermal detection task and placement of optic fibre. LED, light-emitting diode. Middle: example trial structure showing timing of reward window (grey) and the timing of the optical stimulus during trials with optogenetic (opto.) manipulation (blue). Filled circles show licks; first rewarded lick is coloured. Right: response categories of task. stim., stimulation. **b**, Left: raster plot of licks in all trials (n = 300) from an example mouse trained for cooling sorted by stimulus amplitude; green filled circles show false alarms. Right: summary of behavioural performance showing proportion of trials with at least one lick in reward window (n = 5 mice). Grey lines show individual mice; coloured filled circles show mean ± s.e.m. **c**, Same as in **b**, but for warming (raster n = 239 trials, graph n = 6 mice). **d**, Top: latency of first lick is longer in response to warming than to cooling at different amplitudes. Filled circles show data from individual mice in **b**,**c**. Bottom: data from all amplitudes, with filled coloured circles showing mean ± s.e.m. **e**, Top left: proportion of trials with licks in VGAT-ChR2 mice (n = 5 mice) during optical stimulation (cyan) trials versus without optical stimulation (blue) for different cool stimulus amplitudes. Thin grey lines show individual mice; coloured filled circles joined by thick grey lines show mean ± s.e.m. Bottom left: the effect of light stimulus (change in percentage of trials with licks, mean ± s.e.m.) in VGAT-ChR2 mice (yellow), and in mice not expressing ChR2 (grey, see Extended Data Fig. 7). **d**, Right, same as at left, but for warm stimulation (n = 6 mice).

of more neurons as amplitude increases; graded rate coding (Fig. 4a). To address this in the thermal system, we measured single-cell responses in pIC to different amplitude cooling and warming stimuli (Fig. 4b) and show that responses to different amplitudes were reliable within a neuron, but diverse between neurons (Fig. 4c). Plotting the population response amplitude showed an asymmetric response profile with a steeper curve for cooling and more graded for warming (Fig. 4d). Similar data acquired in S1 showed a weak cellular representation of warm amplitudes (Fig. 4d, grey line). In pIC, most cool neurons respond to the smallest amplitude tested (2 °C) with only a minority recruited

panels Fig. 5e and Extended Data Fig. 7), confirming that the light alone does not alter the behaviour of the mice in our task.

We next tested whether the differential impact of optogenetic inhibition of warming and cooling perception resulted from the additional representation of cool in S1. We found that the impact of optogenetic inhibition of S1 is weaker compared to that of pIC (Fig. 5e and Extended Data Fig. 7), supporting the hypothesis that pIC is the central representation of temperature and could be explained by differences in cortical area size, local cellular architecture and projection targets. As expected, simultaneous optogenetic inhibition of S1 and pIC using fibre optics positioned directly over pIC and S1 had a more pronounced effect on warm and cool perception than inhibiting S1 alone (Extended Data Fig. 7), but was similar to the impact of inhibiting pIC alone (Fig. 5e). Notably, even during dual inhibition, mice were still able to detect 10 °C cooling. This could result from the fast onset of cool responses providing a more robust encoding strategy or perhaps reflect the recruitment of different upstream circuits. Future experiments should be designed to address these different hypotheses.

Optogenetic inhibition of pIC in VGAT-ChR2 mice during spontaneous licking of free water rewards and during an acoustic detection task did not alter lick rates (Extended Data Fig. 8). Moreover, optogenetic inhibition of pIC in VGAT-ChR2 mice trained to discriminate between an acoustic stimulus (go; rewarded) versus an acoustic stimulus presented simultaneously with a thermal stimulus (no-go; not rewarded) induced a selective increase of licking during the no-go trials, showing that pIC manipulation has an impact on perception rather than causing a general decrease of licking (Extended Data Fig. 8). Taken together, our behavioural data show that pIC plays a profound role in thermal perception.

## Discussion

Here we identify a cortical region required for non-painful thermal perception and support the hypothesis that pIC houses a thermal cortex. Our data show that humans, monkeys and rodents therefore share not only similar perceptual abilities[18,24,33], but also a cortical area (pIC) specialized in temperature processing, putting the thermosensory system of mammals closer than previously thought[5,34]. Moreover, we observed distinct encoding features of cool and warm in the cortex. The representation of warm has delayed and uniform dynamics that encodes absolute stimulus amplitude, compared to the mixed temporal dynamics for cool that drive a relative encoding of stimulus amplitude. Similar features have been reported in primate and human primary thermal afferent neurons[11–14] and closely resemble thermal response features in *Drosophila* central and peripheral neurons[35,36], together highlighting the conserved nature of thermal encoding across the animal kingdom.

We find that cool and warm are represented in the cortex in a labelled-line-like fashion resembling sensory afferent encoding proprieties (Fig. 1a, top)[9–17]. Nevertheless, cortical neurons can also show response profiles not observed in the periphery, such as the encoding of temperature by thermometer neurons, raising the possibility of complex thermal features being generated in the cortex (Fig. 4c and Extended Data Fig. 5). The presence of distinct features of cool and warm responses both in broadly and tightly tuned pIC neurons shows that functionally segregated streams of afferent information can merge in cortical neurons. This is interesting in light of recent data suggesting interactions between warm and cool afferent pathways[18] and findings from the tactile pathway that suggest substantial subcortical transformation of tactile information before reaching S1 (ref. [37]). Our findings will allow future research studies to examine the separation and integration of different channels of sensory afferent input on the cortical encoding of temperature.

Touch and cool response areas overlap in S1 (ref. [7]), whereas they seem more separate in pIC, hinting at differences both in the sensory input and functional output of these regions. Objects are normally cooler than skin temperature and thermal conductivity is an important component of object identification[38]. One hypothesis is that S1 integrates cool with touch during haptic sensing, whereas pIC encodes the thermal identity (warm versus cool) and level forming an independent and complete representation of thermosensation. The discovery of an optically accessible cortical representation of temperature provides a platform to address the neural mechanisms of non-painful thermal perception, as well as pain evoked during noxious stimulation, allodynia and thermal illusions.

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

## Methods

### Mice

All experimental procedures were carried out in accordance with the State of Berlin Animal Welfare requirements and European animal welfare law. Male and female mice older than 2 months were maintained on a 12:12 h light–dark cycle with experiments carried out during the light phase of the cycle. Mice were housed in groups at $22 \pm 2$ °C temperature and $55 \pm 10$% humidity with ad libitum access to food and water unless stated otherwise. Thy1-GCaMP6s (C57BL/6J-Tg(Thy1-GCaMP6s) GP4.3Dkim/J) mice were used for calcium imaging and for behavioural experiments (The Jackson Laboratory, stock no. 024275)[39,40]. VGAT-ChR2 (B6.Cg-Tg(Slc32a1-COP4*H134R/EYFP)8Gfng/J) mice were used for behavioural experiments (The Jackson Laboratory, stock no. 014548)[41]. For awake experiments, mice were gradually habituated to head and paw fixation and tilted for optical access to pIC or S1.

### Surgery

Mice were deeply anaesthetized (ketamine 120 mg kg$^{-1}$ and xylazine 10 mg kg$^{-1}$) and injected subcutaneously with dexamethasone sodium phosphate (2 mg kg$^{-1}$) to prevent cerebral oedema, as well as metamizol (200 mg kg$^{-1}$). Anaesthetized mice were fixed with a nose-clamp, eye gel (Vidisic, Bausch + Lomb) was applied to both eyes and body temperature was maintained at 37 °C with a heating pad and rectal probe. After surgery, mice received a subcutaneous injection of warm sterile saline solution and were placed on a warm blanket until they awoke from anaesthesia. To avoid post-operative pain, metamizole was dissolved in the drinking water for 3 days post-surgery.

**Implantation of cranial windows.** To implant a window over pIC, mice were anaesthetized as above, and then the parietal and temporal bones of the left hemisphere were exposed and the left temporalis muscle was carefully separated from the temporal bone and partially removed. An approximately $3 \times 3$ mm craniotomy was carried out over pIC as identified by local anatomical landmarks (rhinal vein, middle cerebral artery, zygomatic bone). To implant a window over primary somatosensory forepaw cortex (S1) a 3-mm-diameter craniotomy was carried out over the left hemisphere S1 identified by the anatomical location respect to bregma (1.5–2.5 mm lateral and 0.5–1 mm anterior). To implant windows over pIC and S1 in the same mouse, we carried out two 3-mm craniotomies. The cortical surfaces were then rinsed with Ringer's solution and two glass coverslips were placed on the surface of the cortex. The lower glass (diameter 3 mm or $3 \times 3$ mm) and the upper glass (diameter 4 or 5 mm) were glued together with optical adhesive (NOA 61, Norland Products). Dental cement and cyanoacrylate glue were used to attach a metal head post and the upper coverslip to the skull. Following pIC surgery, the skin lying over the temporal muscle was sutured.

**Clear skull preparation.** To gain simultaneous optical access to S1 and pIC, we used a through-skull preparation (Fig. 1f,g). Under deep isoflurane anaesthesia, the left temporalis muscle was partially removed and the dorsal surface of the skull was cleared of skin and periosteum. Finally, the exposed skull was sealed with cyanoacrylate glue.

### Cortical pharmacology

For pharmacological inactivation, ≈500-μm-diameter craniotomies were drilled over cool (32–22 °C)-responsive areas in S1 and pIC in mice anaesthetized with isoflurane (1.5–2% in $O_2$). Both regions were identified functionally using the widefield calcium imaging response to thermal stimuli. Following the craniotomy, the dura was covered with transparent silicone gel (3-4680, Dow Corning). A 300 nl volume of muscimol or Ringer's solution was injected at a rate of 100 nl min$^{-1}$ 300–500 μm below the pial surface using a pulled glass pipette and a hydraulic injection system (MO-10, Narashige). Muscimol (Abcam, ab120094) was dissolved in Ringer's solution to a concentration of 5 mM. Imaging sessions were carried out >10 min following the end of Ringer's solution or muscimol injection (Fig. 1f,g).

### Sensory stimulation

**Thermal.** For widefield imaging, behavioural and electrophysiological experiments, thermal stimuli were delivered by an $8 \times 8$ mm Peltier element regulated by a feedback-controlled stimulator (Yale Medical School or a custom-made device, ESYS GmbH Berlin). Different thermal stimuli (2-s duration with an onset ramp speed of 20 °C s$^{-1}$) were interleaved randomly and delivered every 30 or 60 s. During paw stimulation, we took care to place the forepaw or hindpaw pad glabrous skin into direct contact with the centre of the Peltier element. During face stimulation, the Peltier surface was positioned on the orofacial region of the snout. For single-cell two-photon experiments, thermal stimuli were delivered using a Peltier-based thermal stimulator (QST-lab). Thermal stimuli of different durations (2, 5, 10 s) or with different onset ramp speeds (approximately 130, 10, 3.3, 2, 1 °C s$^{-1}$) were interleaved randomly and delivered every 30 or 60 s.

**Acoustic.** Acoustic stimuli of 8 kHz, ≈65 dB sound pressure level and 1-s duration were delivered every 60 s using a loud speaker (Visaton) positioned 10 cm from the contralateral ear. The frequency of 8 kHz was chosen because it is well represented in the pIC auditory field[23]. For behavioural training, a 14 -kHz tone (≈65 dB, 1-s duration) was chosen as it is weakly represented in the pIC auditory field[23].

**Vibrotactile.** Vibrotactile stimuli (100 Hz) were generated by a piezo-electric actuator (PL127, PI) equipped with a 5-cm glass rod bent at the tip. The contralateral forepaw or hindpaw was tethered to a rigid support with a hole (diameter 5 mm) to allow the bent tip of the glass rod to stimulate the centre of the palm of the paw. In some experiments vibrotactile stimuli were generated using a small motor (Pololu shaftless vibration motor $8 \times 3.4$ mm) positioned on top of the paw tethered to the Peltier. Vibrotactile stimuli had a 0.5-s duration and were delivered every 30–60 s.

### Widefield imaging

**Setup.** Imaging was carried out with a Leica MZ10F stereomicroscope. Blue excitation light (470 nm) was emitted from a LED (pE-300, CoolLED) and bandpass filtered (470/40 nm). Emission light was bandpass filtered (525/50 nm) before recording with a scientific complementary metal–oxide–semiconductor camera (ORCA-Flash4.O LT, Hamamatsu). Images were acquired at a rate of 20 Hz and 35 ms exposure time. Frame size was 1,024 × 1,024 pixels (2 × 2 binning); for cranial window preparation the field of view was about $2.7 \times 2.7$ mm and for the cleared skull preparation the field of view was about $6.7 \times 6.7$ mm. Acquisition trials lasted 10–15 s and had an inter-trial-interval time of 30–60 s. During pIC imaging, animals were tilted to allow optical access. As a result of physical constraints, pIC and S1 widefield imaging was carried out in different animals and sessions, except during the cortical pharmacological inactivation experiments (Fig. 1f,g). In this case, we used a 'clear skull' preparation and imaged pIC and S1 simultaneously. The setup was always adjusted to allow the paw to be placed in a similar position to all other experiments. Data acquisition was controlled by custom-written code (Python).

**Widefield image analysis.** Videos were motion corrected, and areas in the recordings outside the brain were masked. We calculated the relative change in fluorescence as $\Delta F/F = (F(t) - F_0)/F_0$, in which $F_0$ is the 15th percentile of the activity in the trial. The average activity of the last 500 ms before stimulus was subtracted from $\Delta F/F$. The activity of a given region was quantified by averaging the fluorescence in a region of interest (ROI; diameter 15 pixels) before calculating $\Delta F/F$. When placing ROIs, we avoided visible blood vessels, but otherwise positioned the ROI at the area with the peak stimulus-driven response. The ROI positions

were kept the same over days and recording sessions by comparing the position relative to the blood vessels. Auditory stimuli activated several fields in the auditory cortex and one area in pIC (insular auditory field).

**Somatotopic maps.** Somatotopic maps of sensory input to pIC (Fig. 1) and S1 (Extended Data Fig. 2) are grand averages across mice and trials. Not all stimuli were delivered to all mice. Using the ROI of the thermal forepaw response as a reference location, $\Delta F/F$ videos for all trials were aligned by translation. The grand averages were smoothed with Gaussian filter ($\sigma = 20$ pixels) and thresholded to show only peak activity (>85% of the maximum activity for thermal and >90% for tactile stimulation). For pIC, because of the simultaneous responses in the auditory insular field and auditory cortex to sound stimuli, responses were analysed independently. We carried out this analysis only on data with grand average peak response amplitude >3% $\Delta F/F$.

**Response onset estimation.** We defined the onset of the widefield response as the peak of the second derivative of the widefield signal in a time window from the start of stimulus to the peak of the first derivative. If negative values of the first derivative appeared after stimulus start, the time window was from the last of such negative values. We carried out this analysis only on data with a peak response amplitude >3% $\Delta F/F$.

## Two-photon imaging

**Setup.** Fluorescence signals were recorded using a two-photon microscope (ThorLabs Bergamo II, 12 kHz scanner) with a Nikon 16× water-immersion objective (NA 0.8) giving a field of view of 540 × 540 µm. The scope was operated using ThorImageLS software (v4.0.2019.8191, Thorlabs). In most experiments the microscope was rotated by 45° from vertical. Owing to physical constraints, pIC and S1 two-photon imaging was carried out in different animals and sessions. The excitation laser (InSight DeepSee, Spectra-Physics) was tuned to 940 nm, and the power never exceeded 100 mW. Emitted photons were bandpass filtered 525/50 (green) onto a GaAsP photomultiplier tube. Multi-plane (512 × 512 pixels) acquisition was controlled by a fast piezoelectric objective scanner, with planes spaced 45 µm apart in depth. Seven planes were acquired sequentially, and the scanning of the entire stack was repeated at about 5 Hz. Beam turnarounds at the edges of the image were blanked. Acquisition trials lasted 26 s and had an inter-trial interval of 30–60 s. Stimuli generation and hardware synchronization were carried out on a computer with a National Instrument card running custom-written Python code.

**Two-photon analysis.** Motion correction of data, identification of putative neurons and calculation of $\Delta F/F$ was carried out using the Suite2p package (v0.9.3) in Python[42]. Identified neurons were manually verified by visual examination of the traces and the spatial footprints of the neurons. A Savitzky–Golay filter was applied to traces presented in figures for visual purpose alone.

**Criteria for responsive neurons.** Neurons with a significant increase in activity to 10 °C cooling and warming stimuli were included for further analysis. Cool and warm neurons were defined as those significantly responding to either 10 °C cooling or warming stimuli. Broadly tuned neurons had significant response to both cooling and warming stimuli. To identify an increase in response amplitude, the activity in a time window before stimulus onset was compared with the activity in a time window during stimulus. A neuron was identified as being responsive if the change in activity was significantly larger (estimated by bootstrapping) than two times the noise level of the neuron measured by the interquartile range of activity before the stimuli.

The thermal bias index (Fig. 2d,e) for a thermally responding neuron is the normalized difference between its cool and warm responses, (warm − cool)/(warm + cool), in which 'warm' is the average peak response to 32–42 °C and 'cool' is the average peak response to 32–22 °C.

Grand average (Fig. 3b,c) calculated from all significantly responding neurons shown in Fig. 2 and includes 470 neurons with significant responses to cooling and 401 to warming. Of these, 125 responded to both. The duration index (Fig. 3d) measures the change in the fluorescence level between the initial peak value ($f_{init}$) and the end ($f_{end}$) of a 2-s thermal stimulus, ($f_{init} - f_{end}$)/$f_{init}$. The adaptation index (Fig. 3g) was calculated as the difference in the maximum fluorescence level during fast thermal stimulation (about 130 °C s$^{-1}$; $f_{fast}$) and during slow thermal stimulation (≈1 °C s$^{-1}$; $f_{slow}$), ($f_{fast} - f_{slow}$)/$f_{fast}$.

## Behavioural experiments

**Setup.** Head-fixed mice (VGAT-ChR2 and Thy1-GCaMP6s) were implanted with a glass window over pIC and trained on a go/no-go stimulus detection paradigm. Mice were rewarded with a water drop (about 2 µl) if they reported a randomly timed thermal stimulus with at least one lick of a water spout (capacitance sensor) within a window of opportunity (from the start of thermal stimulus to start of offset ramp). Correct rejections were not rewarded and incorrect responses were not punished, although premature licking in the 5 s before the stimulus onset would postpone the next trial by 5 s. Mice were water restricted and their weight was monitored daily. For behavioural training, a 200-µm-diameter, 0.22-NA optic fibre (Thorlabs) was coupled to an LED (470 Plexon LED source) and placed on the coverslip surface orthogonal to the thermal region of the pIC or S1 using the blood vessel pattern. In a subset of mice, cranial windows were implanted above both S1 and pIC, and LEDs were placed above both areas (Extended Data Fig. 7). Control of behavioural training and data collection was carried out using either custom-written Labview software (16.0f5, National Instruments, USA) or the Bpod system (1.8.2, Sanworks, USA).

**Thermal perception task.** Thermal perception (Fig. 5) training involved several stages: (1) Free access to water rewards from the water spout; (2) automatic water rewards paired to the presentation of 10 °C, 2-s cooling and warming stimuli from AT 32 °C; (3) training to report a 10 °C cooling and warming stimuli by licking for a water reward within a 2 s window from the start to the end of the plateau phase of the stimulus; thermal stimuli trials and catch trials were presented with a ratio of 1:1 with an inter-stimulus-interval of 15–20 s; (4) once a hit rate >70% and false alarm rate <30% had been reached, 4 randomized stimulus amplitudes were used from AT 32 °C (cooling, 22 °C, 30 °C, 31 °C and 31.5 °C; warming 42 °C, 40 °C, 38 °C and 34 °C) with a 4:2 ratio of stimulus trials to catch trials; (5) once a hit rate of >70% and false alarm rate <30% had been achieved for at least 1 amplitude, in the next session we presented the same 4 thermal stimuli with 1 catch trial with LED on or off (on/off trials) with a ratio of 2:1 (randomized). The LED was on for the entire duration of the stimulus and delivered at 20 Hz, 50% duty cycle with a power of 12.5 mW (measured at tip of fibre). Before the start of stage (4) or (5) sessions, mice were exposed to a brief session of the stage (3) protocol to quench initial thirst.

**Thermal discrimination task.** The thermal discrimination task training (Extended Data Fig. 8) involved several stages: (1) free access to water rewards from the water spout; (2) automatic water rewards paired to the presentation of 14-kHz, ≈65-dB, 0.5-s acoustic stimulus; (3) training to report 14-kHz stimulus by licking for a water reward within a 1.4-s window from the start of the stimulus (go trial); acoustic stimuli (14 kHz, about 65 dB) delivered 0.6 s after the beginning of a 2-s thermal stimulus (either cooling or warming) (no-go trial) and catch trials (no stimuli) were not rewarded; go, no-go and catch trials were presented with a ratio of 1:1:1 with an inter-stimulus-interval of 15–20 s; training was continued until go trial hit rate >70%, no-go false alarm rate around 50% and false alarm catch rate <30%; (4) once a mouse learned to discriminate between acoustic stimulus alone (go) and acoustic stimulus presented together with thermal stimuli (no-go), in the next session we presented the same stimuli interleaved at a ratio of 2:1 into trials with

the LED off (off) and trials with the LED on (on). The LED was on for the entire duration of the thermal stimulus and delivered at 20 Hz, 50% duty cycle with a power of 12.5 mW (measured at tip of fibre). Before the start of stage (4) sessions, mice were exposed to a brief session of the stage (3) protocol to quench initial thirst.

**Acoustic perception.** The behaviour training to report acoustic stimuli (Extended Data Fig. 8) had the same structure as the thermal training above, but used a 14-kHz, ≈65-dB, 2-s acoustic stimulus.

**Free licking.** To monitor the impact of pIC optogenetic manipulation on licking behaviour, water restricted mice were allowed to freely lick from a lick spout with continuous rewards. Licking rates during the 2 s before the onset of the light stimulus (Off) were compared with those during the 2 s after the onset of the light stimulus (On). Light stimuli were delivered for 5 s at 12.5 mW, 20 Hz and 50% duty cycle (Extended Data Fig. 8).

**Datasets and analysis.** Mice were randomly allocated to be trained first to report cooling or warming. In a second training round, they were trained to report the opposite stimulus. Data measuring the thermal perceptual ability of mice without optogenetic manipulation (Fig. 5b–d) were generated from the same mice used in optogenetic manipulations (Fig. 5e,f), the day before optogenetic testing. The effect of the optogenetic manipulation (bottom panels of Fig. 5e,f and Extended Data Fig. 7) was quantified by the mean change in the percentage of lick trials.

## In vivo electrophysiology

Electrophysiological recordings for measuring the impact of optogenetic inhibition on thermal responses were carried out in VGAT-ChR2 mice implanted with a head post. Mice were habituated for head and paw fixation. On the day of the recording, the mouse was briefly anaesthetized (about 30 min, 1% isoflurane) and a craniotomy of about 2 mm diameter was carried out over the pIC. The dura was removed in the ventral part of the craniotomy and the brain was covered with Kwik-Cast silicon (World Precision Instruments). The animal was placed in its home cage to recover from anaesthesia for at least 2–3 h before head fixation in the recording setup.

We carried out 7 recordings in 4 mice using 32-channel silicon probes (Neuronexus) in either a linear or four-shank 'Buzsaki32' configuration. In some experiments, the tips of the probe were covered with DiI (Sigma). The probe was inserted orthogonal to the pIC with a micromanipulator (Luigs&Neumann) at 1–2 μm s⁻¹. Recordings were acquired at 30 kHz with a Neuralynx system (Cheetah). The extracellular recordings were spike sorted using Kilosort (version 2)[43]. The data were collected to evaluate the effect of ChR2 on neural activity, so we carried out limited manual sorting to exclude obvious noise but did not take any steps to exclude multi-unit activity. Units with a mean activity during stimulus representation exceeding a threshold of 2 s.d. above the background were considered to be thermally responsive.

Two-second-long thermal stimuli of 10 °C cooling and warming were interleaved randomly and delivered every 30 s. The LED fibre (200-μm-diameter, 0.22-NA optic fibre, Thorlabs) was tilted slightly to accommodate the silicone probe. The LED was on for the entire duration of the thermal stimulus and delivered at 20 Hz, 50% duty cycle with a power of 12.5 mW (measured at tip of fibre).

## Histology

To label the thermal cortical representation, animals were deeply anaesthetized by intraperitoneal injection of ketamine and xylazine. The centre of the response area was marked with a glass pipette painted with fluorescent dye (DiI, 5 mg ml⁻¹). Mice were then perfused transcardially using cold phosphate-buffered saline (PBS; 0.1 M, pH 7.4) and fixed with paraformaldehyde (PFA; 2% in 0.1 M PBS). The brain was removed

and kept for 1–5 h in 2% PFA. After washing with PBS, the hemispheres were separated and cortices were removed and flattened between two glass slides separated by a spacer of 1–2 mm. Glass slides were weighed down for approximately 3–8 h at 4 °C in 2% PFA. After washing with PBS, 70–80-μm sections were cut on a Vibratome (Leica VT1000s). Sections were stained for cytochrome oxidase activity (2 mg cytochrome c, 6 mg diaminobenzidine). After the staining procedure, sections were mounted on glass slides with Mowiol mounting medium. Images were acquired with a Zeiss microscope (AX10) using a 5× objective and processed using Fiji (ImageJ, NIH, USA). Area borders were manually delineated following the contrast of the cytochrome oxidase stain and are comparable to previously reported data[44]. Coronal sections of 50 μm in thickness of PFA-fixed mouse brains were stained for 48 h with the following primary antibodies: mouse anti-Gad67 (catalogue number MAB5406; clone 1G10.2; Millipore; 1:800); chicken anti-GFP (catalogue number ab13970; Abcam; 1:250); mouse anti-NeuN (catalogue number MAB377; clone A60; Millipore; 1:100). The secondary antibodies Cy3 goat anti-mouse (A-21422; Invitrogen; 1:250) and A488 goat anti-chicken (A-11039; Invitrogen; 1:250) were incubated for a few hours at room temperature. Cell counting was carried out manually in Fiji using the Cell Counter plug-in on epifluorescence images acquired using a Zeiss microscope (AX10) using a 10× objective.

## Data analysis and statistics

No statistical methods were used to predetermine sample sizes, but our sample sizes were similar to those used in previous publications. Experimenters were not blind to trial order, but trial order was randomized during the experiment. All data analysis was carried out using Python. Uncertainty of means are reported with standard error of the mean. Bootstrapping was used for estimating central 95% confidence intervals using 5,000 resamples. Some histograms in Fig. 3 and Extended Data Fig. 4 do not show outlier data points. In Fig. 1f,g, we used two-sided paired $t$-test. For Fig. 1f: cool S1, $t = -3.78$, $P = 0.0323$; warm S1, $t = -0.17$, $P = 0.8767$; cool pIC, $t = 0.44$, $P = 0.6929$; warm pIC, $t = -0.45$, $P = 0.6839$. For Fig. 1g: cool S1, $t = 0.55$, $P = 0.6182$; warm S1, $t = -0.68$, $P = 0.5454$; cool pIC, $t = -3.61$, $P = 0.0366$; warm pIC, $t = -8.52$, $P = 0.0034$.

## Reporting summary

Further information on research design is available in the Nature Portfolio Reporting Summary linked to this article.

## Data availability

Data are available from the corresponding authors upon request. Source data are provided with this paper.

## Code availability

Code is available from the corresponding authors upon request.

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

**Acknowledgements** This work was supported by the European Research Council (ERC-2015-CoG-682422, J.F.A.P.), the European Union (3x3Dimaging 323945, J.F.A.P.), the Deutsche Forschungsgemeinschaft (FOR 2143, J.F.A.P.; SFB 1315, J.F.A.P.), the Helmholtz Society (J.F.A.P.), the Centre National de la Recherche Scientifique (M.C.), the Independent Research Fund Denmark (M.V.) and the MDC-Weizmann Helmholtz International Research School iNAMES (G.G.). We thank M. Brecht, G. Lewin, J. Kremkow, S. Crochet and members of the laboratory of J.F.A.P. for constructive comments on an earlier version of the manuscript, S. Steinfelder for help with administrative and technical aspects, J. König for help with immunohistochemistry, B. Eickholt for sharing NeuN antibody and A. Burkhalter for advice on the flattened cortex preparation and cortical parcellation.

**Author contributions** M.V., M.C. and J.F.A.P. designed the study. M.V. M.C. and G.G. carried out experiments and analysed the data. M.V., M.C. and J.F.A.P. wrote the manuscript.

**Funding** Open access funding provided by Max-Delbrück-Centrum für Molekulare Medizin in der Helmholtz-Gemeinschaft (MDC).

**Competing interests** The authors declare no competing interests.

**Additional information**
**Correspondence and requests for materials** should be addressed to M. Vestergaard, M. Carta or J. F. A. Poulet.

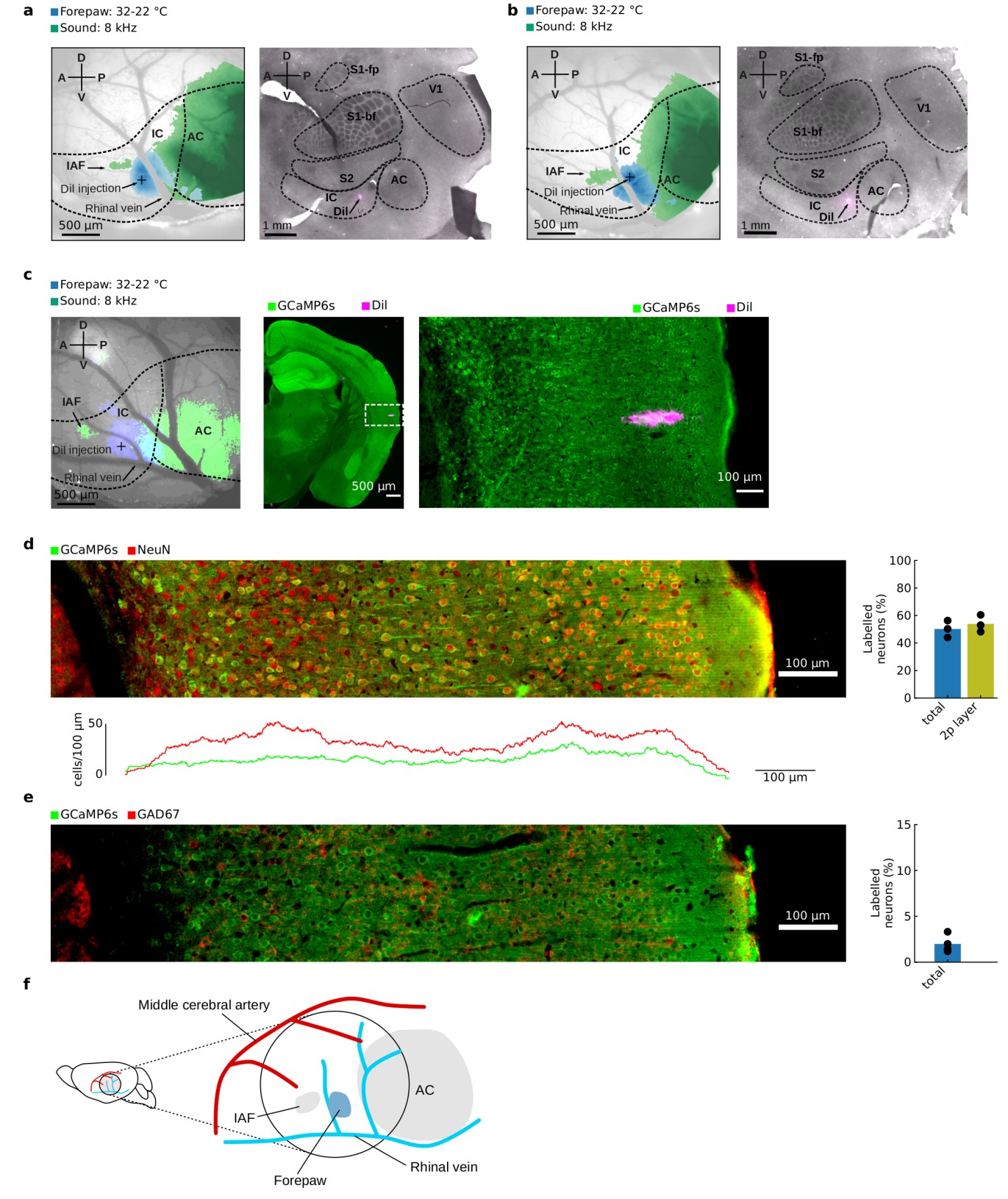

**Extended Data Fig. 1** | See next page for caption.

**Extended Data Fig. 1 | Functional and histological localization of pIC.**
**a**, Functional and histological localization of the thermal zone of forepaw pIC in 2 example mice. Left, in vivo image of cortical surface of a Thy1-GCaMP6s mouse with superimposed, coloured areas showing the widefield responses to cool (blue, 32-22 °C) and sound (green, 8 kHz). Glass pipette painted with DiI (5 mg/ml) was inserted in the center of the cool response area (black cross). Right, cytochrome oxidase staining of a flattened cortex preparation from the same mouse on left with site of DiI injection (magenta). Areal borders were manually delineated based on CO intensity reaction, parcellation followed[44]. **b**, Second example mouse as in **a**. **c**, Left, in vivo image of cortical surface of a Thy1-GCaMP6s mouse with superimposed, coloured areas showing the widefield responses to cool (blue, 32-22 °C) and sound (green, 8 kHz). Center, coronal section of the same mouse on the left containing the site of DiI injection (magenta). GCaMP6s signal was amplified with immunohistochemistry with anti-GFP antibody. Right, zoom of the injection site. Example mouse is the same presented in Fig. 2c. Similar results were found in other two mice. **d**, Top, example of epifluorescence image of pIC from a Thy1-GCaMP6s mouse stained for GCaMP6s (labelled with anti-GFP antibody) and for the general neuronal marker NeuN. Bottom, cell density of NeuN- and GCaMP-positive neurons. Right, quantification of the percentage of NeuN-positive neurons that also expressing GCaMP6s in the imaged section across layers (total) and in the imaging layer (2p, 100–370 µm) (n = 3 mice, mean). Values are in line with previous data[40]. Filled circles show data from different mice. **e**, Same as **d**, but with staining for GCaMP6s and with antibody for GABAergic neurons (GAD67). Right, percentage of GAD67-positive neurons among GCaMP6s-expressing neurons (n = 4 mice, mean). Values are in line with previous data[45]. **f**, Cartoon schematic representing the position of the acoustic and forepaw cool responsive fields relative to major blood vessels. IAF, insular auditory field; AC, auditory cortex. Different terminology has been used for this region in previous studies, including pIC, parietal ventral area (PV) or S2/IC, highlighting the difficulties in discerning two neighboring cortical regions. In agreement with studies that have used functional mapping of sensory responses[22,23,29,46,47] and because pIC is a region associated with thermal processing in human studies[2,4,5,31,48,49] here we use the term pIC.

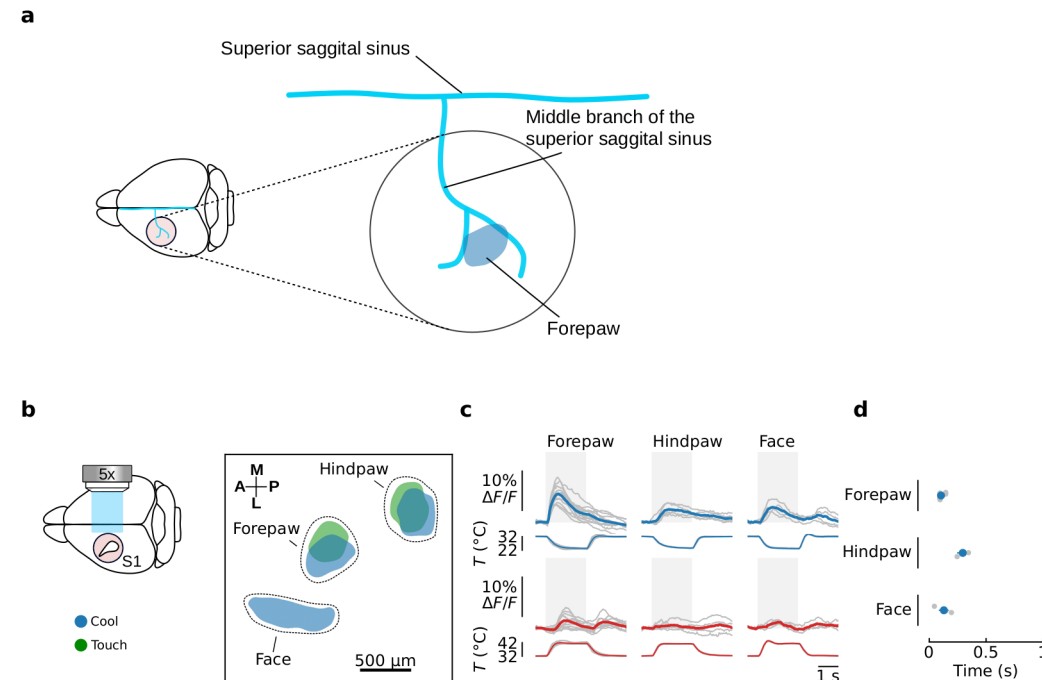

**Extended Data Fig. 2 | Somatotopic representation of temperature in S1.**
**a**, Cartoon schematic showing the location of the forepaw cool responsive area (blue) relative to the major blood vessels. **b**, From left to right, cartoon schematic showing imaging setup; somatotopic map of responses locations to thermal (10 °C cool) and tactile. Coloured area indicates peak population response averaged across mice (see methods, *n* = 12 thermal forepaw, 5 thermal hindpaw, 4 thermal face, 12 touch forepaw, 5 touch hindpaw). Data from individual mice are aligned to peak activity of the thermal forepaw response. **c**, Widefield responses to thermal stimulation of different body parts from the same dataset as in **b**. Grey lines show mean responses from individual mice (*n* = same as **b**), coloured lines show population mean, grey area indicates time from start of stimulus to end of plateau phase. **d**, Grey filled circles show response latencies from individual mice, coloured filled circles show mean ± s.e.m., same data as in **c**.

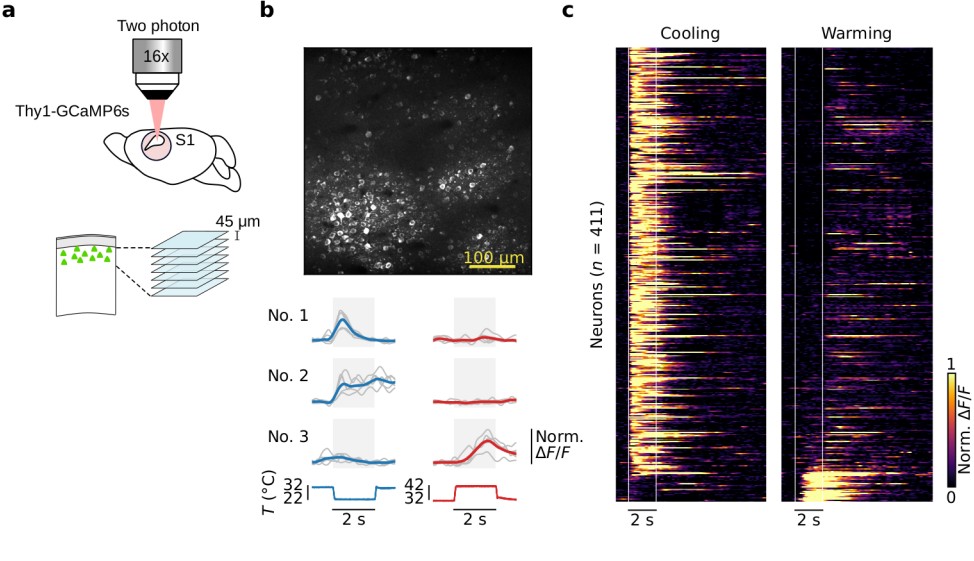

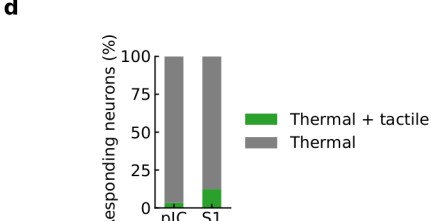

**Extended Data Fig. 3 | Cellular encoding of temperature in S1. a,** Cartoon schematic showing two-photon imaging of S1. Imaging started at 100 µm from the pial surface of a Thy1-GCaMP6s mouse, 7 optical sections were acquired with intervals of 45 µm. **b,** Top, example in vivo two photon image of GCaMP6 expressing neurons in S1. Bottom, responses of three example S1 neurons during 10 °C cooling (blue) or 10 °C warming (red) stimuli. Here and in all figures, grey lines show single trial responses, coloured lines show average, grey shaded box indicates time from start of stimulus to end of plateau phase. Below, corresponding thermal stimulus traces. **c,** Single neuron S1 calcium responses to cooling (left) and warming (right) stimuli. Each line represents a

single neuron, responses are normalized to the peak and sorted based on the thermal bias index with white lines showing the onset and the end of the plateau phase of thermal stimuli (n = 411 neurons, 4 mice, 9 sessions). **d,** Bar graph showing the percentage of pIC and S1 neurons responding to thermal only and also to tactile stimulation (pIC, 506 neurons thermal only, 18 neurons thermal and tactile; S1, 345 neurons thermal only, 49 neurons thermal and tactile). The overall number and proportion of thermal and tactile neurons in S1 presented here differs from a previous report[50]. This might depend on the body part studied, stimulus design and recording conditions.

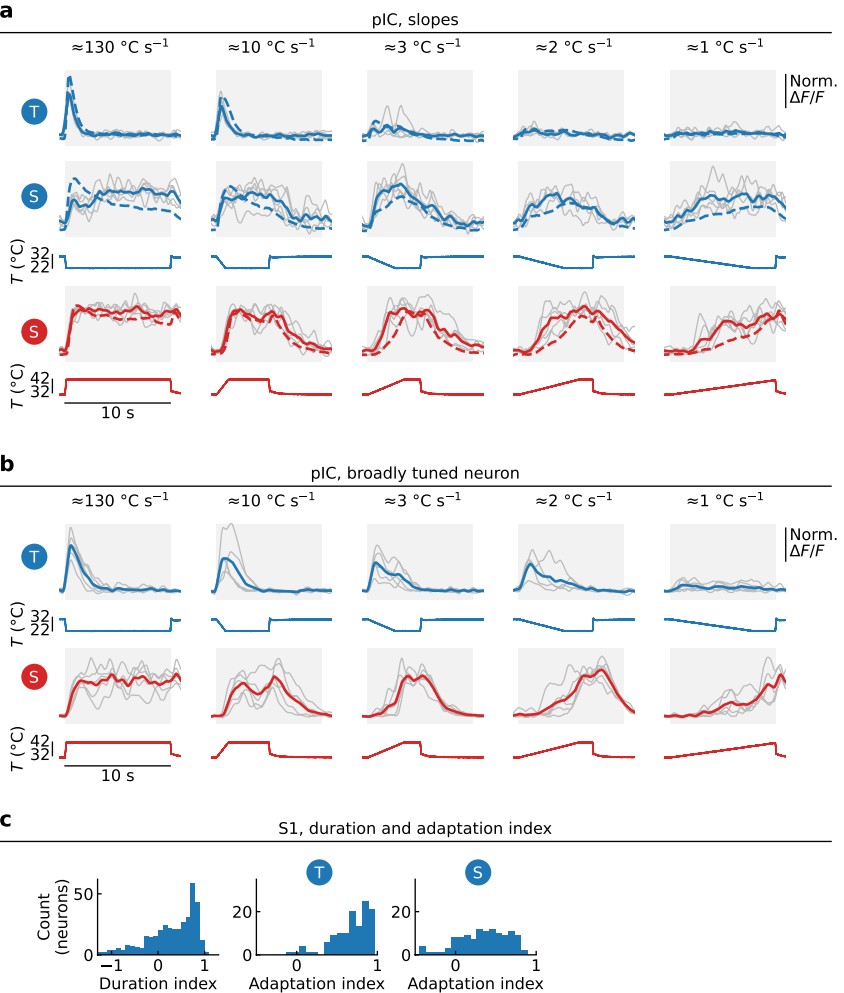

**Extended Data Fig. 4 | Dynamics of thermal responses to thermal stimuli with different stimulus onset speeds. a**, Responses of example transient and sustained cool (T and S) and sustained warm (S) neurons responding to stimuli with different onset speeds of approximately 130, 10, 3, 2, 1 °Cs⁻¹. Example neurons are the same as presented in main Fig. 3g. Grey lines show individual trials, coloured lines show mean response. Dashed lines show mean response of the corresponding neuronal population. **b**, Responses of example pIC broadly tuned neuron responding to the same stimuli used in **a**. **c**, Duration and adaptation index for cool neurons in S1 (*n* = 122 cool transient neurons, 118 cool sustained neurons, 3 mice, 7 sessions).

**a**

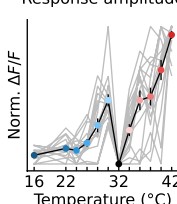

Response amplitude

Norm. ΔF/F

16  22    32    42
Temperature (°C)

**b**

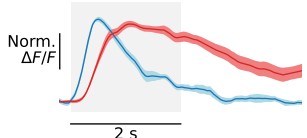

Norm.
ΔF/F

2 s

**Extended Data Fig. 5 | Thermometer neurons. a**, Two-photon calcium imaging response amplitude plotted as a function of the thermal stimulus for neurons with similar response profile to neuron #5 of Fig. 4c (*n* = 16 thermometer neurons from 411 warm neurons in Fig. 4; ~5%). Grey lines show data for individual neurons, coloured filled circles show mean ± s.e.m. at AT 32 °C. Most thermometer neurons are characterized by low threshold response to warming stimuli and had cool responses only for small amplitude cooling stimuli. **b**, Grand average responses (mean ± s.e.m.) of thermometer neurons to 2 °C cooling (32-30 °C, blue) or 10 °C warming (32–42 °C, red) from AT 32 °C. Thermometer neurons conserve the latency difference as seen for cool and warm responses in tightly tuned and broadly tuned neurons (Fig. 3b, c). The latency of thermal responses in these neurons indicate that they receive the same input as the tightly tuned and broadly tuned neurons. These neurons highlight complex thermal features possibly constructed by the pIC. One hypothesis is that the lack of responses to large cooling stimuli could be explain by disynaptic, feedback, inhibition driven by a secondary cool channel activated by high amplitude cooling stimuli (eg. 32-24 °C).

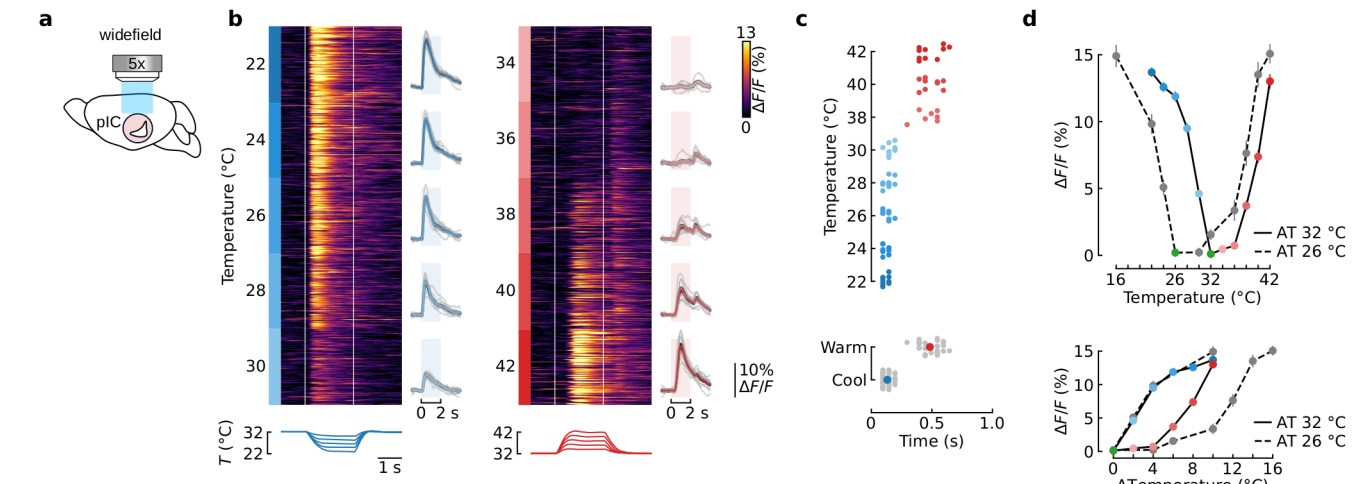

**Extended Data Fig. 6 | Widefield imaging analysis of thermal amplitude encoding in pIC. a**, Left, schematic of widefield calcium imaging in a mouse implanted with a cranial window over pIC. **b**, Widefield responses of pIC to forepaw thermal stimulation with different amplitudes (*n* = 123–128 trials per temperature, 10 mice). Heat maps showing widefield responses to thermal stimuli with each line representing a single trial response and thermal stimuli below with white lines showing the onset and the end of the plateau phase of thermal stimuli; coloured lines to the right of heatmap show mean ± s.e.m. across trials from all mice and grey lines show mean response from individual mice; coloured shaded boxes indicate start of thermal stimulus and end of plateau phase. Right, same as left panels but for warm responses. **c**, Top,

response latencies to different amplitudes of thermal stimuli from individual mice. Bottom, grey filled circles show data from all amplitudes (*n* = 10 mice). Filled circles show mean ± s.e.m. **d**, Top, peak population response to different amplitude stimuli from experiments presented in **b**. Coloured filled circles with connecting lines show data with AT of 32 °C. Grey filled circles with connecting dashed lines show data using AT of 26 °C (filled circles show mean ± s.e.m) (*n* = 6 mice). Green filled circles shows activity at AT. Bottom, same data as in top graph but showing population response amplitude plotted against thermal stimulus amplitude. Graph shows shift in warm response amplitude but not for cool when presented with a lower AT.

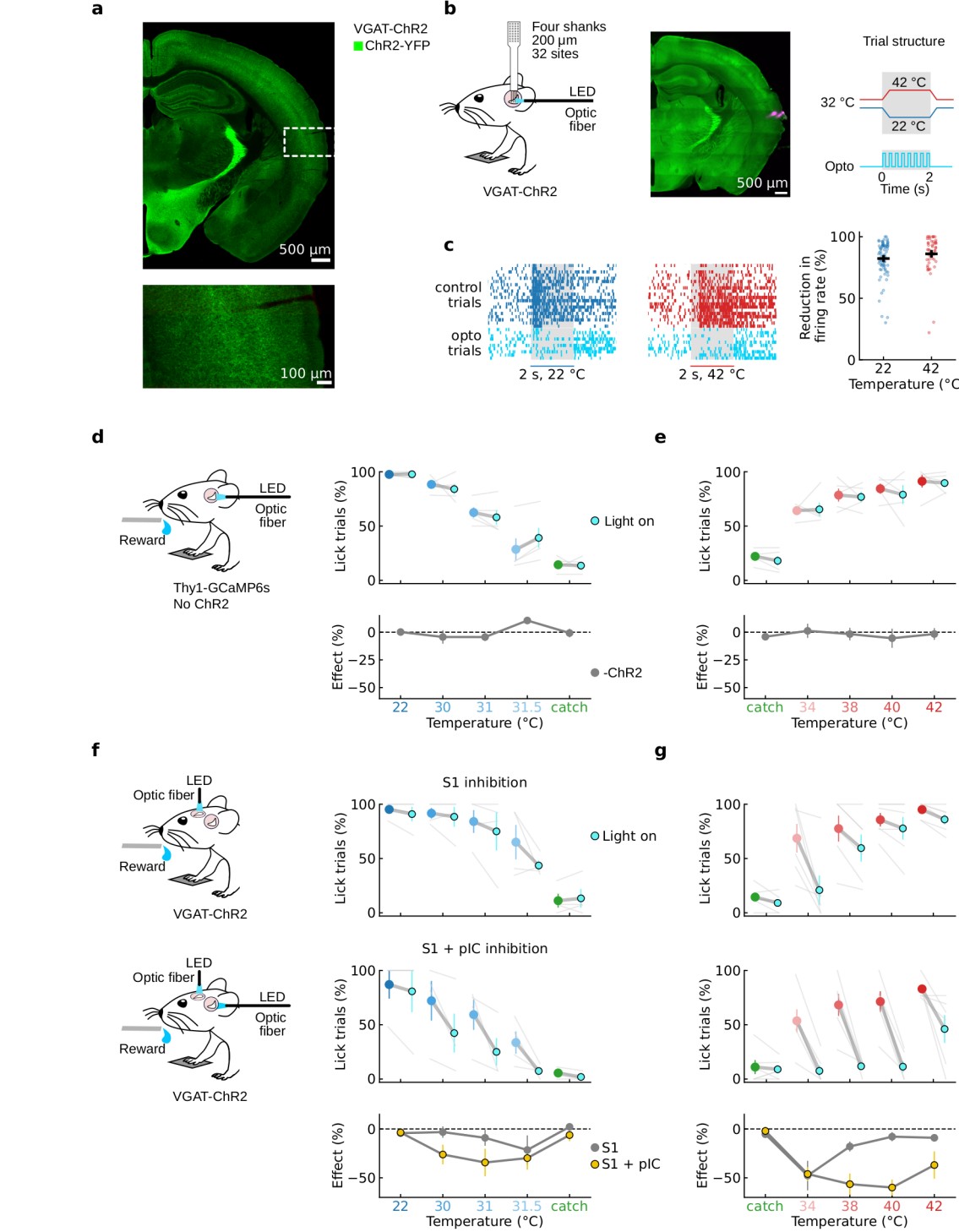

**Extended Data Fig. 7 |** See next page for caption.

**Extended Data Fig. 7 | Electrophysiological validation of optogenetic inhibition and behavioural controls on S1 optogenetic silencing. a**, Example coronal slices from VGAT-ChR2 mice co-labelled with anti-GFP antibody showing homogeneous expression across pIC. Similar results were found in two other mice. **b**, Left, schematic showing an awake VGAT-ChR2 mouse with right forepaw on a Peltier element during electrophysiological recording and optogenetic manipulation of left pIC. Center, example coronal section of a VGAT-ChR2 mouse showing 2 tracks of the 4 shanks electrode in magenta. Similar results were found in another mouse. Right, trial structure showing the timing of the optical stimulation stimulus relative to the thermal stimulus. **c**, Left, raster plot of example unit responses to 20 trials of cool (blue) or warm (red) stimulation and during thermal stimulation during optogenetic silencing of pIC (10 trials, cyan). Grey background box indicates timing of stimulus presentation for cool (blue) and warm (red). Pulsed LED on for the entire duration of the thermal stimulus. Thermal stimuli presentation was randomized. Trials with pIC optogenetic inhibition were also randomized, but are separated here for clarity. Right, shows graph of percentage reduction in firing rate of excitatory units during optogenetic stimulus trials in 4 mice, blue filed circles show 79 cold units and red shows 53 warm unit, black bars show mean ± s.e.m. **d**, Optical stimulation of the thermal zone of pIC in mice that do not express ChR2 (Thy1-GCaMP6s) does not affect cool perception. Behavioural task was identical to that presented in Fig. 5. Top graph shows number of trials with a lick during optical stimulation (cyan) vs. without optical stimulation (blue or red) at different thermal stimulus amplitudes, green shows catch trials. Grey thin lines show individual mice ($n$ = 5), coloured filled circles connected by thick grey thick lines shows mean ± s.e.m. Lower panel shows the effect of light stimulus as the change in percentage of trials with licks, (mean ± s.e.m) corresponding to the stimulus amplitude above. **e**, same as in **d** but for warm. **f**, Top, proportion of trials with licks in VGAT-ChR2 mice ($n$ = 4) during optical stimulation of S1 (cyan) vs. trials without optical stimulation (blue or red) for different cool stimulus amplitudes green shows catch trials. Grey lines show individual mice, coloured filled circles connected by thick grey lines show mean ± s.e.m. Middle, same as top but for simultaneous optogenetic inhibition of S1 and pIC. Bottom, shows the effect of light stimulus (change in percentage of trials with licks, mean ± s.e.m.) from mice with optogenetic inhibition of S1 alone (yellow filled circles) or from mice with optogenetic inhibition of both S1 and pIC simultaneously. **g**, Same as **e**, but for warm stimulation ($n$ = 6 mice). S1 inactivation caused a smaller effect on cool perception in comparison to our previous report[7], however here we used a small diameter optical fiber for transient optogenetic inhibition, as compared to prior pharmacological inhibition.

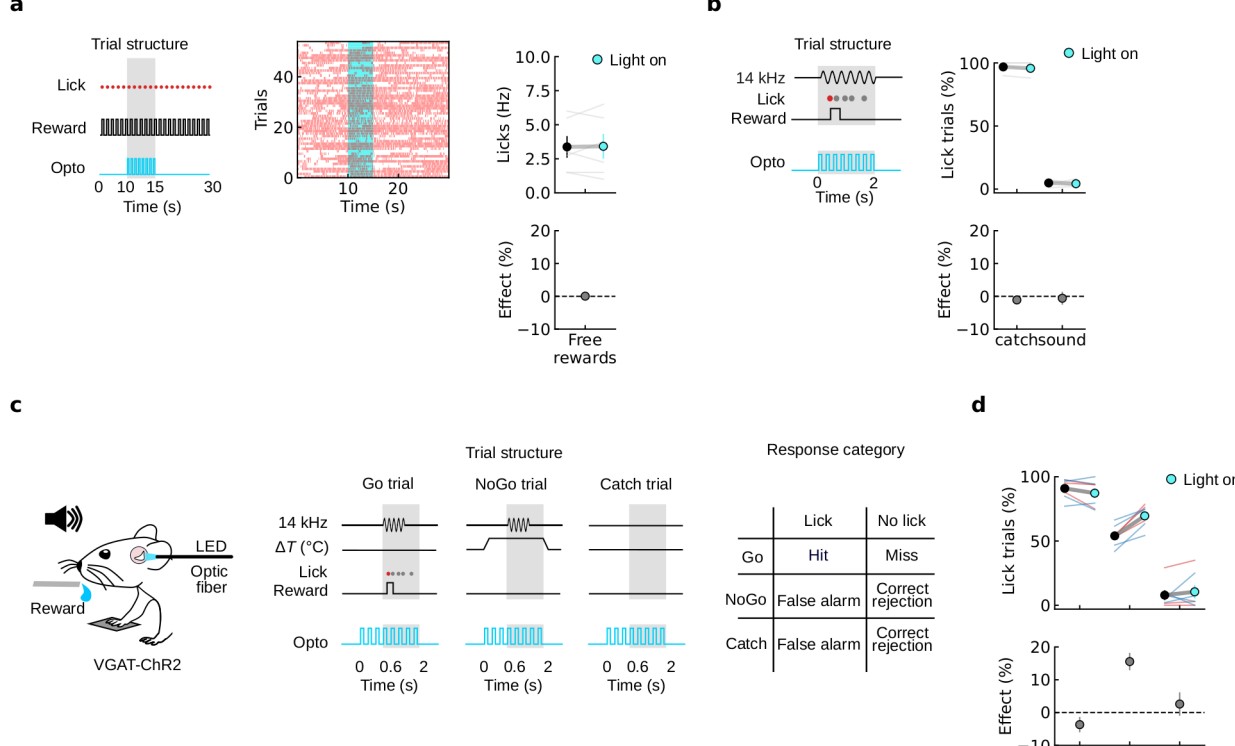

**Extended Data Fig. 8 | Behavioural control experiments. a**, Optogenetic inhibition of pIC using the VGAT-ChR2 mice does not affect spontaneous licking behaviour. Left, trial structure showing optical stimulation during free licking (grey bar). Middle, example raster plot of free licking rates in a VGAT-ChR2 mouse during optical stimulation. Right, mean lick rates in VGAT-ChR2 mice during optical stimulation Off (black; 2 s before light stimulus) vs. On (cyan; 2 s beginning of light stimulus) trials. Grey thin lines show individual mice ($n = 6$), coloured filled circles connected by thick grey lines show mean ± s.e.m. **b**, Auditory perception is not altered by optogenetic inhibition of pIC. Perception task design is same as thermal task but uses an acoustic stimulus (14 kHz, 65 dB SPL). Left, structure of optical stimulation trial. Right, mean lick rates in VGAT-ChR2 mice during trials with optical stimulation during the auditory stimuli (cyan) vs. trials without optical stimulation (black). Grey thin lines show individual mice ($n = 4$), coloured filled circles connected by thick grey thick lines shows mean ± s.e.m. **c**, Left, schematic of discrimination task and placement of the optic fiber. Middle, example trial structure showing timing of reward window (grey) and the timing of an optical stimulus during trials with optogenetic manipulation. Filled circles show licks, first lick coloured to show rewarded lick. VGAT-ChR2 mice were trained to discriminate a 14 kHz acoustic stimulus (Go trial) from the same stimulus presented 0.6 s after the beginning of a 2 s thermal stimulus (cooling or warming) (NoGo trial). Right, response categories of the task. Go trials were rewarded, NoGo and catch trials (no stimuli) were not rewarded. Training continued until Go trial hit rate > 70%, NoGo false alarm rate ~50% and false alarm rate < 30%. A criterion of ~50% for the NoGo false alarm rate was chosen to be able to observe an increase or decrease in lick rates during optogenetic inhibition. The amplitude of the thermal stimuli (cooling or warming) was adjusted from mouse to mouse to obtain this criterion. **d**, Top, proportion of trials with licks in VGAT-ChR2 mice during optical stimulation (cyan) trials vs. without optical stimulation (black). Blue and red lines show data from individual mice trained with either cooling (blue, $n = 4$) or warming (red, $n = 3$) stimuli. Coloured filled circles joined by grey thick lines show mean ± s.e.m. Bottom, shows the effect of light stimulus (change in percentage of trials with licks, mean ± s.e.m.). The increase, rather than decrease, in NoGo false alarm rate during optogenetic manipulation confirms the direct and selective involvement of pIC in thermosensory processing.

# Reporting Summary

## Statistics

For all statistical analyses, confirm that the following items are present in the figure legend, table legend, main text, or Methods section.

| n/a | Confirmed | |
|---|---|---|
| ☐ | ☒ | The exact sample size (*n*) for each experimental group/condition, given as a discrete number and unit of measurement |
| ☐ | ☒ | A statement on whether measurements were taken from distinct samples or whether the same sample was measured repeatedly |
| ☐ | ☒ | The statistical test(s) used AND whether they are one- or two-sided *Only common tests should be described solely by name; describe more complex techniques in the Methods section.* |
| ☒ | ☐ | A description of all covariates tested |
| ☐ | ☒ | A description of any assumptions or corrections, such as tests of normality and adjustment for multiple comparisons |
| ☐ | ☒ | A full description of the statistical parameters including central tendency (e.g. means) or other basic estimates (e.g. regression coefficient) AND variation (e.g. standard deviation) or associated estimates of uncertainty (e.g. confidence intervals) |
| ☐ | ☒ | For null hypothesis testing, the test statistic (e.g. *F*, *t*, *r*) with confidence intervals, effect sizes, degrees of freedom and *P* value noted *Give P values as exact values whenever suitable.* |
| ☒ | ☐ | For Bayesian analysis, information on the choice of priors and Markov chain Monte Carlo settings |
| ☒ | ☐ | For hierarchical and complex designs, identification of the appropriate level for tests and full reporting of outcomes |
| ☐ | ☒ | Estimates of effect sizes (e.g. Cohen's *d*, Pearson's *r*), indicating how they were calculated |

*Our web collection on statistics for biologists contains articles on many of the points above.*

## Software and code

Policy information about availability of computer code

| Data collection | For running behaviour experiments a custom script was created in Labview (16.0f5, National Instruments, USA) or with the Bpod system (1.8.2, Sanworks, USA). A script was created in Python for widefield and two photon imaging together with the use of ThorImageLS software (v4.0.2019.8191). Neuralynx (Cheetah) was used for electrophysiology data. |
|---|---|
| Data analysis | Motion correction of two photon data, identification of putative neurons and calculation of ΔF/F was done using the Suite2p package (v0.9.3) for Python (Pachitariu et al BioRxiv 2017). The extracellular recordings were spike sorted using Kilosort (version, 2)(Pachitariu et al BioRxiv 2016). A script was created in Python to analyze imaging and behavior data using standard techniques described in the methods. |

For manuscripts utilizing custom algorithms or software that are central to the research but not yet described in published literature, software must be made available to editors and reviewers. We strongly encourage code deposition in a community repository (e.g. GitHub). See the Nature Portfolio guidelines for submitting code & software for further information.

## Data

Policy information about availability of data

All manuscripts must include a data availability statement. This statement should provide the following information, where applicable:

- Accession codes, unique identifiers, or web links for publicly available datasets
- A description of any restrictions on data availability
- For clinical datasets or third party data, please ensure that the statement adheres to our policy

Data are available from the corresponding author upon reasonable request.

## Human research participants

Policy information about studies involving human research participants and Sex and Gender in Research.

Reporting on sex and gender
*Use the terms sex (biological attribute) and gender (shaped by social and cultural circumstances) carefully in order to avoid confusing both terms. Indicate if findings apply to only one sex or gender; describe whether sex and gender were considered in study design whether sex and/or gender was determined based on self-reporting or assigned and methods used. Provide in the source data disaggregated sex and gender data where this information has been collected, and consent has been obtained for sharing of individual-level data; provide overall numbers in this Reporting Summary. Please state if this information has not been collected. Report sex- and gender-based analyses where performed, justify reasons for lack of sex- and gender-based analysis.*

Population characteristics
*Describe the covariate-relevant population characteristics of the human research participants (e.g. age, genotypic information, past and current diagnosis and treatment categories). If you filled out the behavioural & social sciences study design questions and have nothing to add here, write "See above."*

Recruitment
*Describe how participants were recruited. Outline any potential self-selection bias or other biases that may be present and how these are likely to impact results.*

Ethics oversight
*Identify the organization(s) that approved the study protocol.*

Note that full information on the approval of the study protocol must also be provided in the manuscript.

# Field-specific reporting

Please select the one below that is the best fit for your research. If you are not sure, read the appropriate sections before making your selection.

☒ Life sciences        ☐ Behavioural & social sciences        ☐ Ecological, evolutionary & environmental sciences

For a reference copy of the document with all sections, see nature.com/documents/nr-reporting-summary-flat.pdf

# Life sciences study design

All studies must disclose on these points even when the disclosure is negative.

Sample size
Sample sizes were not predetermined based on statistical methods but on sizes described in recent literature and experience from previous studies (Milenkovic et al., Nat Neuroscience 2014; Chet et al., Current biology 2021; Chet et al., Science 2011; Ohki et al., Nature, 2005, Peng et al., Nature, 2015).

Data exclusions
Imaging experiments: In a few cases imaging sessions were excluded due to poor optical access, movement or very low number of trials. Behavioral experiments: No data were excluded from animals able to reach a minimal performance on the detection task (>70% correct and <30% false alarm).

Replication
All attempts of replication were successful. Imaging data were replicated in several cohorts of mice. Behavioral data was replicated in two cohorts.

Randomization
Imaging experiments: Allocation to different groups was random. Stimulation temperatures given were in general interleaved. Behavioral experiments: Allocation to different groups was random.

Blinding
Experimenter was not blind to the genotype of the animal. The initial phase the analysis of behavior data was done blind to the genotype.

# Reporting for specific materials, systems and methods

We require information from authors about some types of materials, experimental systems and methods used in many studies. Here, indicate whether each material, system or method listed is relevant to your study. If you are not sure if a list item applies to your research, read the appropriate section before selecting a response.

## Materials & experimental systems

| n/a | Involved in the study |
|---|---|
| ☐ | ☒ Antibodies |
| ☒ | ☐ Eukaryotic cell lines |
| ☒ | ☐ Palaeontology and archaeology |
| ☐ | ☒ Animals and other organisms |
| ☒ | ☐ Clinical data |
| ☒ | ☐ Dual use research of concern |

## Methods

| n/a | Involved in the study |
|---|---|
| ☒ | ☐ ChIP-seq |
| ☒ | ☐ Flow cytometry |
| ☒ | ☐ MRI-based neuroimaging |

# Antibodies

| Antibodies used | mouse anti-Gad67 (cat. no.: clone 1G10.2, #MAB5406; Millipore; 1:800)<br>chicken anti-GFP (cat. no.: ab13970; Abcam; 1:250)<br>mouse anti-NeuN (cat. no.: clone A60 #MAB377; Millipore; 1:100)<br>Cy3 goat anti mouse (A-21422; Invitrogen; 1:250)<br>A488 goat anti chicken (A-11039; Invitrogen; 1:250) |
|---|---|
| Validation | Antibodies were only used if validated by the manufacturer via their website. All antibodies used are standard in the field.<br><br>All antibodies used in this study were validated as described a the following websites (and refrences therein):<br>hhttps://www.merckmillipore.com/FR/fr/product/Anti-GAD67-Antibody-clone-1G10.2,MM_NF-MAB5406;<br>https://www.abcam.com/gfp-antibody-ab13970.html;<br>https://www.merckmillipore.com/FR/fr/product/Anti-NeuN-Antibody-clone-A60,MM_NF-MAB377;<br>https://www.thermofisher.com/antibody/product/Goat-anti-Mouse-IgG-H-L-Cross-Adsorbed-Secondary-Antibody-Polyclonal/A-21422;<br>https://www.thermofisher.com/antibody/product/Goat-anti-Chicken-IgY-H-L-Secondary-Antibody-Polyclonal/A-11039. |

# Animals and other research organisms

Policy information about studies involving animals; ARRIVE guidelines recommended for reporting animal research, and Sex and Gender in Research

| Laboratory animals | In the study male and female mice >2 month old of the the following strains were used: Thy1-GCaMP mice (C57BL/6J-Tg(Thy1-GCaMP6s)GP4.3Dkim/J) and VGAT-CHR2 (B6.Cg-Tg(Slc32a1-COP4*H134R/EYFP)8Gfng/J. They were kept in a 12:12 hour dark:light cycle with experiments performed during the light phase of the cycle. Mice were housed in groups at 22-24°C temperature and 45-55% humidity with ad libitum access to food and water unless stated. |
|---|---|
| Wild animals | No wild animals were used in the study |
| Reporting on sex | We used 26 male and 39 female mice and did not select mice for each experiment based on sex. Because the study was not designed to identify sex differences in thermal encoding, our sample sizes do not allow meaningful analysis of sex differences. |
| Field-collected samples | No field collected samples were used in the study. |
| Ethics oversight | Approved licenses from Landesamt für Gesundheit und Soziales (LAGeSo), Berlin, Germany |

Note that full information on the approval of the study protocol must also be provided in the manuscript.

