## [Peer Review File · Nature]

Manuscript Title: The cellular coding of temperature in the mammalian cortex

Redactions – unpublished data

Reviewer Comments & Author Rebuttals

Reviewer Reports on the Initial Version:

Referee #1 (Remarks to the Author):

The study by Vestergaard et al. addresses the cortical correlates of cooling and warming sensations in mice. The authors employ a combination of wide-field and two-photon calcium imaging to investigate temperature coding in S1 as well as the posterior insular cortex (pIC). The authors demonstrate that while S1 is only encoding cooling, the pIC processes cooling and warming. Finally, the authors use inhibition of neural activity in the pIC to show a necessary role for the pIC in thermal perception.

The study is well designed and very interesting. The techniques employed are state of the art and the manuscript is very clearly written. I have a only a few comments that could potentially strengthen the study:

- 1) The authors convincingly demonstrate somatotopic organization of thermal representations in pIC, however their analyses of S1 thermal responses is restricted to the forepaw. It would, in my opinion, be a great addition and important for a comprehensive comparison of thermal coding properties of S1 and pIC to also address hindpaw and face cooling / warming in S1 and their relationship to potential somatosensory representations of these body parts.
- 2) The authors use Thy1-GCamP6s mice for all their calcium imaging. Since Thy1 is not per se a comprehensive marker for all cortical excitatory neurons, in my opinion, it would be nice to see the level of expression of GCamP in their mouse line, first by raw images of the imaging fields counterstained with a pyramidal cell marker and a quantification of the fraction of pyramidal cells expressing GCamP6s. Without this data, it is difficult to judge whether the results obtained are specific to a subpopulation of pyramidal neurons or a general property of all excitatory neurons in the cortical regions imaged.
- 3) More details and controls for the optogenetic inhibition experiments should be provided: please demonstrate that activation of VGAT+ cells successfully inhibits the activity of excitatory neurons at the durations and parameters used. Would the effect of the cooling experiments be larger if the authors inhibited S1 and pIC simultaneously?

Referee #2 (Remarks to the Author):

How temperature stimuli are represented in the cortex has been extensively studied in many animal models including human. Microstimulation and lesion studies in both animal and human supported important role of both S1 and pIC in encoding temperature information. However, detailed analysis at single cell level in rodent model is still missing. In this study, the authors performed in vivo two photon calcium imaging examined representation of temperature stimulus in both SI and pIC. They found that both SI and pIC shown robust responses to cooling but only pIC response to warming.

They further examined the kinetics of cortical temperature responding neurons and found that they are similar to response properties reported in sensory neurons and in spinal cord neurons.

While the studies provide new cellular data support a critical role of the pIC in encoding temperature information, there are several concerns with regard to the premise, novelty, and conclusions.

Major points:

1. The authors indicate in the abstract that the somatosensory cortex does not represent warming. While the difference between S1 and insular cortex is big, it is clear that the S1 also responds to warming (Extended Data Fig. 3c). Moreover, this claim is not consistent with a recent study using similar two-photon in vivo calcium imaging method, but reported robust responses to warm stimulus in S1 of awake mice (Chen XJ. et al., 2021 Journal of Neuroscience). In that study, heat stimuli was delivered using a infrared laser emitter onto the rostral back skin. The laser intensity was calibrated to heating up the skin to 40 celsius degree, which is 2 degree lower than the current study.
2. The result that the insular cortex uses separate maps for touch and temperature while the S1 has a bigger overlap is interesting but preliminary. The maps are not drawn systematically (3 body areas in the insular cortex and 1 area in the S1, Extended Data Fig. 2). The authors should at least add 2 areas to the S1 data to see if the representation of forepaw cooling is closer to touching the same body area or cooling of a different area. Moreover, Extended Data Fig. 3d shows only 14% of S1 neurons respond to both temperature and tactile stimuli. This number is much lower than reported by Chen XJ. et al., 2021 Journal of Neuroscience, which shows 43% of tactile responding neurons also response to heating.
3. Data in Fig. 5 (optogenetics) demonstrated critical role of the pIC in temperature guided behavior. But the authors only did insula but not S1. Previous publication from the same lab have already shown pharmacological inactivation of forepaw S1 can prevent cooling perception (Milenkovic et al., 2014 Nature Neuroscience). I am curious if optogenetic silencing the S1 would also impact the behavior. if so, I think S1 should not just be a redundant circuit as claimed in current manuscript.

Minor points:

1. The author reply quite a bit on example neurons. Figures like Extended Data Fig. 4 only have 1 neuron each. It would be better if they show all recorded neurons.
2. It is unclear if insular imaging and S1 imaging are performed in the same mice during the same session. If not, that would severely affect the conclusion as I expect different paw tethering during the different imaging sessions would result in a variability of ~1 degree.

Referee #3 (Remarks to the Author):

This study investigates how temperature is encoded in mammalian cortex. The authors show using widefield calcium imaging that posterior insula (pIC) responds to both warming and cooling, whereas S1 responds only to cooling. Two-photon imaging shows that most pIC neurons are biased toward either warm or cold, and that the responses properties of these neurons resemble primary sensory afferents (e.g. warm responses are more sustained and encode absolute temperature, whereas cold

responses rapidly adapt and encode change). The functional significance of these findings is demonstrated by optogenetically silencing pIC and showing that mice have impaired ability to report warming and cooling of the paw.

This is an interesting and timely paper that reports fundamental new information about how temperature perception is encoded in insular cortex. The experiments appear logically designed and carefully executed and the paper is clearly written. The finding that distinct pIC neurons encode warm and cold with response properties that match labelled line models from peripheral neurons is exciting.

My only major criticism is that the authors do not perform any experiments that directly test how manipulation of primary afferents or their thermosensitive channels alters the encoding of temperature in pIC. The authors' recent work (Paricio-Montesinos et al, *Neuron*, 2020) argues that temperature encoding in the periphery is not a simple labelled line defined by thermoTRPs but instead that TRPM8 mediates both cold and warm responses (in part via a warm induced suppression of a tonically active TRPM8+ population). Given that there is still uncertainty about how these various channels contribute to sensing of warmth, it would significantly elevate the manuscript to perform imaging in pIC in TRPM8 KO mice (or following similar manipulations) in order to test some of these models at the level of cortical dynamics.

Reply to Referees

Nature

Cellular encoding of temperature in the mammalian cortex

Mikkel Vestergaard, Mario Carta, Gamze Güney, and James F.A. Poulet

Comments from Referees are in black Arial font.

Authors' responses and draft text are in cyan Arial font.

We thank the Referees for their reading of our manuscript, their enthusiasm for our findings and helpful and constructive comments. We have addressed the Referees' comments with new experiments, extra analysis as well as changes to the text. The major changes include:

- Analysis of GCaMP6 expression across pIC and imaging layer (Extended Data Fig. 1).
- Widefield imaging to characterize the thermal somatotopy within S1 across different body parts (Extended Data Fig. 2).
- Testing of the impact of optogenetic inhibition of pIC on neuronal responses to thermal stimulation (Extended Data Fig. 7)
- Determining the impact of S1 on thermal perception using optogenetic inhibition in trained mice (Extended Data Fig. 7)
- Additional analysis of S1 cellular encoding (Fig. 4d, Extended Data Fig. 4c)
- Presentation of population averaged data during encoding of different ramp speeds (Extended Data Fig. 4a)
- Discussion of the role of S1 and sensory afferent channels in cortical encoding and perception of temperature (Discussion)

Referee #1

The study by Vestergaard et al. addresses the cortical correlates of cooling and warming sensations in mice. The authors employ a combination of wide-field and two-photon calcium imaging to investigate temperature coding in S1 as well as the posterior insular cortex (pIC). The authors demonstrate that while S1 is only encoding cooling, the pIC processes cooling and warming. Finally, the authors use inhibition of neural activity in the pIC to show a necessary role for the pIC in thermal perception. The study is well designed and very interesting. The techniques employed are state of the art and the manuscript is very clearly written. I have a only a few comments that could potentially strengthen the study:

We thank the Referee for their supportive comments.

1) The authors convincingly demonstrate somatotopic organization of thermal representations in pIC, however their analyses of S1 thermal responses is restricted to the forepaw. It would, in my opinion, be a great addition and important for a comprehensive comparison of thermal coding properties of S1 and pIC to also address hindpaw and face cooling / warming in S1 and their relationship to potential somatosensory representations of these body parts.

Thank you for raising this point we agree that the comparative somatotopy in pIC and S1 is an interesting result. To address this more comprehensively, we used the same experimental settings to those used for pIC (Fig. 1h-j and see Methods) but implanted a cranial window centered over forepaw S1. As for pIC (Fig. 1), during S1 imaging we stimulated the hindpaw and forepaw and facial region with thermal stimuli (warm/cool) and the hindpaw and forepaw with vibrotactile stimuli. Facial tactile stimulation was avoided for both S1 and pIC imaging because it required a more specific and gentle approach to obtain clear response, most likely due to high sensitivity of facial area and movements of the face in an awake animal.

We found that:

- (i) Vibrotactile stimulation of the forepaw and hindpaw elicited spatially separated cortical responses in line with known somatotopic maps of S1 (Gămănuț et al., 2018).
- (ii) Cool responses for forepaw, hindpaw and face overlap with their respective tactile response area or with known maps (confirming Milenkovic et al. 2014).
- (iii) For all 3 body parts tested we did not observe reliable warm responses in S1.

The data are shown below in Referee Fig. 1 and as Extended Data Fig. 2.

Referee Figure 1. Somatotopic representation of temperature in forepaw S1. **b**, from left to right, cartoon schematic showing imaging setup; somatotopic map of responses locations to thermal (10°C cool) and tactile stimuli. Coloured area indicates the peak population response averaged across mice (see Methods, $n = 12$ thermal forepaw, 5 thermal hindpaw, 4 thermal face, 12 touch forepaw, 5 touch hindpaw, $n =$ mice). Data from individual mice are aligned to peak activity of the thermal forepaw response. **c**, Widefield responses to thermal stimulation of different body parts from the same dataset as in **b**. Grey lines show mean responses from individual mice (n values same as **b**), coloured lines show population mean, and grey area indicates time from start of thermal stimulus to end of plateau phase. **d**, Grey filled circles show response latencies from individual mice, coloured filled circles show mean \pm s.e.m., same data as in **c**.

The larger amplitude forepaw S1 responses to cool compared to the hindpaw and face is interesting and could reflect differences in afferent drive but an understanding of the mechanism requires further experiments. Although in the hindpaw and facial region we did not observe a response to warming in S1, small amplitude, variable signals were observed in some forepaw datasets. These weak and unreliable responses agree well with our two-photon imaging data that shows only a small minority of neurons with a variable and long latency response during warm stimulation. Because of their variability and low amplitude, they could result from a number of different non-sensory sources. Another possibility is that they result from presynaptic axonal activity from warm responsive cells in pIC that project their axons to S1. Future work could address these possibilities.

Overall, these data support our finding that S1 is primarily driven by cooling. Mice use their forepaws for haptic sensing and objects in the natural environment are typically cooler than skin temperature. We hypothesize that the temperature representation in S1 is required for the fast detection and formation of a coherent somatosensory percept during haptic sensing. We have now changed the wording in the Results section:

Manuscript L79-80: 'Intriguingly, while responses to cooling and touch spatially overlap in S1 (Extended Data Fig. 2)⁷, they appear more separate in pIC (Fig. 1h).'

To address the Referee's request for a more comprehensive comparison of thermal coding properties of S1 and pIC, we also include new single cell analysis of S1 thermal responses, including:

- (i) Response dynamic analysis for cooling with different slopes (Extended Data Fig. 4).
- (ii) Encoding of cool and warm amplitude by S1 neurons (Fig. 4d, grey line).

These data show that the principles of cooling encoding are similar between S1 and pIC and highlight the weak warm responsiveness in S1. These data have mentioned this in the Results:

Manuscript L171-172: 'Similar results were observed for S1 cool responses (Extended Data Fig. 4).'

Manuscript L184-186: 'These features were observed in highly tuned and broadly tuned pIC cells and also in S1 cool neurons (Extended Data Fig. 4).'

Manuscript L214-215: 'Similar data acquired in S1 showed a weak cellular representation of warm amplitudes (Fig. 4d, grey line).'

2) The authors use Thy1-GCamP6s mice for all their calcium imaging. Since Thy1 is not per se a comprehensive marker for all cortical excitatory neurons, in my opinion, it would be nice to see the level of expression of GCamP in their mouse line, first by raw images of the imaging fields

counterstained with a pyramidal cell marker and a quantification of the fraction of pyramidal cells expressing GCaMP6s. Without this data, it is difficult to judge whether the results obtained are specific to a subpopulation of pyramidal neurons or a general property of all excitatory neurons in the cortical regions imaged.

Referee 1 correctly points out that Thy1 expression can be restricted to subsets of cortical excitatory neurons. For this reason, we decided to use the mouse line GP4.3 (Dana et al., 2014a), that offers a widespread, homogeneous expression of GCaMP6s in L2/3 and L5 across different cortical regions. In the GP4.3 line (as in many other Thy1 lines) L4 granular neurons do not express GCaMP6s (Dana et al., 2014). Unfortunately, pyramidal cell marking antibodies are notoriously variable and it is difficult to identify one that reliably labels all pyramidal neurons in a manner suitable for quantitative measurements of cell numbers. Instead, co-labelling of NeuN and GCaMP is a standard approach (Dana et al., 2014; Scott et al., 2018). Therefore, to address the expression of GCaMP6 in our imaging planes of pIC more thoroughly, we have performed new analysis examining how the numbers of NeuN+ and GCaMP6+ neurons change with distance from the pial surface (Extended Data Fig. 1). In agreement with the original study that first characterized this mouse line (Dana et al., 2014), our analysis confirms that around 50-60% of cells in the imaging layer (100-370 μm from pial surface) expressed GCaMP6. The data are shown below (Referee Fig. 2) and as Extended Data Fig. 1.

Referee Figure 2. Expression of GCaMP6s in the pIC of the GP4.3 mouse line. **d**, Top, example of epifluorescence image of pIC from a Thy1-GCaMP6s mouse stained for GCaMP6s (labelled with anti-GFP antibody) and for the general neuronal marker NeuN. Bottom, cell density of NeuN- and GCaMP- positive neurons. Right, quantification of the percentage of NeuN-positive neurons that also expressing GCaMP6s in the imaged section across layers (total) and in the imaging layer (2p, 100-370 μm) ($n = 3$ mice, mean). Values are in line with previous reports (Dana et al., 2014b). Filled circles show data from different mice.

These data show that in the GP4.3 line there is no strong evidence for selective expression of GCaMP in pyramidal subtypes in pIC in this line because:

- (i) GCaMP is widely expressed across cortical layers;
- (ii) the percentage of labelled L2/3 neurons in pIC are high (about 50-60%) and similar to the original study that examined this mouse line in V1, S1 and M1 and that concluded a broad expression of GCaMP across L2/3 (Dana et al., 2014);
- (iii) few GABA-ergic interneurons are labelled (about 2%, Extended Data Fig. 1) (Makino et al., 2017).

The GP4.3 line has been extensively used by many laboratories investigating various aspects of cortical processing and broad expression of GCaMP is observed across different cortical regions (Dana et al., 2014a; Hennestad et al., 2021; Liu et al., 2019; Makino et al., 2017; Romero et al., 2020) and so far, no reports suggest that a subpopulation of pyramidal neurons is differentially expressing GCaMP in this line.

3) More details and controls for the optogenetic inhibition experiments should be provided: please demonstrate that activation of VGAT⁺ cells successfully inhibits the activity of excitatory neurons at the durations and parameters used.

We agree that showing the direct impact of VGAT-ChR2 stimulation as a method of silencing cortex is an important control experiment. To address this, we performed extracellular electrophysiological recordings from the pIC superficial layers of awake VGAT-ChR2 mice during thermal stimulation of the paw and optogenetic stimulation via an optic fiber. Stimulus trials were interleaved with and without light. The light was presented simultaneously with the stimulus. Light and stimulus parameters were as close as possible to those used for the behavioural experiments and described in the Methods section of the Supplementary Information.

Supplementary Information L187-191: 'For behavioural training, a 200 μm diameter, 0.22 NA optic fibre (Thorlabs) was coupled to an LED (470 Plexon LED source) and placed on the coverslip surface orthogonal to the thermal region of the pIC or S1 using the blood vessel pattern. In a subset of mice, cranial windows were implanted both in top of S1 and pIC and LEDs were placed both in top of both areas (Extended Data Fig. 7).'

Supplementary Information L206-208: 'The LED was on for the entire duration of the stimulus and delivered at 20 Hz, 50% duty cycle with a power of 12.5 mW (measured at tip of fiber).'

The data are shown below (Referee Fig. 3) and as Extended Data Fig. 7.

Referee Figure 3. Optogenetic inhibition of thermal responses in pIC. **b** Left, schematic showing an awake VGAT-ChR2 mouse with right forepaw on a Peltier element during electrophysiological recording and optogenetic manipulation of left pIC. Center, example coronal section of a VGAT-ChR2 mouse showing 2 tracks of the 4 shanks electrode (magenta). Right, trial structure showing the timing of the optical stimulus synchronous with the thermal stimulus. **c**, Left, raster plot of example unit responses to cool or warm stimulation (20 trials each; blue and red) and during stimulation while silencing of pIC (10 trials each, cyan). Blue and red horizontal lines are periods of stimulus presentation for cool and warm respectively. LED as on for the entire duration of the thermal stimulus. Stimuli were presented in random order. Trials with pIC optogenetic inhibition were also randomized but are separated here for clarity. Right, reduction in firing rate of excitatory units during opto stimulus trials (79 cold and 53 warm units in 4 mice).

Activity from multi-electrode recordings showed reliable responses to cool and warm stimuli that were efficiently inhibited by optogenetic stimulation of the pIC. Across the population we observed similar impact of optogenetic inhibition on warm responses and cool responses (inhibition of cool

= 82%; inhibition warm = 86%). Together, our data show that optogenetic inhibition of pIC efficiently suppresses thermal responses in pIC.

Would the effect of the cooling experiments be larger if the authors inhibited S1 and pIC simultaneously?

Thank you for raising this interesting experiment. One hypothesis is that the cool representation in S1 provides additional information that allows the mouse to sense cooling even when pIC is inhibited. To address the Referee's question, and assess the role of S1 in thermal perception further, we performed new optogenetic manipulations of S1 and pIC in VGAT-ChR2 mice trained to detect warm or cool of different amplitude (Referee Fig. 4, Extended Data Fig. 7).

Referee Figure 4. Optogenetic inhibition of S1 and pIC alters thermal perception. **f**, Top, proportion of trials with licks in VGAT-ChR2 mice ($n = 4$) during optical stimulation (cyan) of S1 trials vs. without optical stimulation (blue) for different cool stimulus amplitudes. Grey lines show individual mice, coloured filled circles show mean \pm s.e.m. Middle, same as top row but for simultaneous photoinhibition of both S1 and pIC. Bottom, shows the effect of light stimulus (change in percentage of trials with licks, mean \pm s.e.m.) from mice with photoinhibition of S1 alone (yellow) or from mice with photoinhibition of both S1 and pIC simultaneously. **g**, Same as **f**, but for warm stimulation ($n = 6$ mice).

We optogenetically inhibited S1 using the same optic fiber and light stimulation parameters as for our original pIC experiments and observed a weak inhibitory impact of S1 inhibition on cool and warm detection for low amplitudes (0.5°C cooling, 2°C warming). A decreased perception of cool detection during S1 inhibition was expected, however the impact appears smaller than in our prior study (Milenkovic et al., 2014). We think this might be because previously we used pharmacological inhibition of S1 that likely had a more significant and longer lasting inhibitory effect on S1 compared to transient optogenetic inhibition presented only during the stimulus period.

The impact of S1 inhibition on cool and warm perception is also weaker compared to the same experiment in pIC (Fig. 5). We think this is because pIC is the primary thermal cortex and has a more powerful role on thermal perception than S1. To keep our experimental protocols as similar as possible we used the same diameter fiber optic for optogenetic stimulation of S1 and pIC it

should be noted that S1 is a larger cortical area than pIC and could contain a different local architecture, ratio of cell types and projection targets. Overall, a more detailed, mechanistic understanding of the comparative impact of optogenetic inhibition between S1 vs pIC will require further detailed experiments in future studies.

Given the extremely small numbers of cells with excitatory responses to warm in S1, the inhibitory impact of S1 inhibition on low amplitude warm perception seemed counterintuitive at first thought. However, S1 optogenetic inhibition could induce many different sensory effects which have a secondary impact on perception. For example, it might alter the perception of the hand size or shape. Moreover, pIC and S1 are reciprocally connected (Gămănuț et al., 2018; Gehrlach et al., 2020, Bokinić et al. bioRxiv 2022), therefore inhibiting one area could influence sensory encoding in the other. Again, it difficult to interpret these results further without more detailed experiments addressing different and specific hypotheses.

Finally, to address the Referee's question directly, we performed simultaneous optogenetic inhibition of S1 and pIC in trained mice using two fiber optics, one positioned directly over pIC and the other over S1. As expected, this had a far more pronounced effect on warm and cool perception than inhibiting S1 alone. Notably, however, even during dual inhibition, mice were still able to detect 10°C cooling. Our electrophysiological results above (Referee Fig. 3) show that the majority of recorded neurons are inhibited by the optogenetic inhibition, however optogenetic inhibition does not completely inhibit all spikes evoked by thermal stimuli and one possibility is that the rapid and short latency spikes evoked by cooling make the cool percept robust and resistant to manipulations and drive downstream circuits involved in perception. Previous studies have shown that subcortical circuits can also play a central role in the detection of salient, simple stimuli (Ceballo et al., 2019; Hong et al., 2018). We have observed that cool is particularly salient for mice, they are able to learn to report cooling even in the first training session and the sensory threshold is low (Paricio-Montesinos et al., 2020), therefore another possibility is that subcortical circuits, unaffected by cortical VGAT inhibition, play a major role in the detection of the more salient cool stimulus compared to warm. These hypotheses could be tested with future studies.

These data are presented in the main text.

Manuscript L281-292: 'We next tested whether the differential impact of optogenetic inhibition to warming and cooling perception resulted from the additional representation of cool in S1. We found that the impact of optogenetic inhibition of S1 is weaker compared to pIC (Extended Data Fig. 7, Fig. 5e and f), supporting the hypothesis that pIC is the central representation of temperature and could be explained by differences in cortical area size, local cellular architecture as well as projection targets. As expected, simultaneous optogenetic inhibition of S1 and pIC using fiber optics positioned directly over pIC and S1 had a more pronounced effect on warm and cool perception than inhibiting S1 alone (Extended Data Fig. 7), but was similar to the impact of inhibiting pIC alone (Fig. 5e and f). Notably, even during dual inhibition, mice were still able to detect 10°C cooling. This could result from the fast onset of cool responses providing a more robust encoding strategy or perhaps reflect the recruitment of different upstream circuits. Future experiments should be designed to address these different hypotheses.'

The relevance of the new data is mentioned in the Discussion:

Manuscript L345-347: 'One hypothesis is that the cool-only representation in S1 integrates cool with touch during haptic sensing, whereas pIC encodes the thermal sign (warm vs. cool) and level forming an independent and complete representation of thermosensation.'

References

- Bokiniec, P, Whitmire C. J., Leva T. M., Poulet J. F.A. (2022) Brain-wide connectivity map of mouse thermosensory cortices. bioRxiv. doi: <https://doi.org/10.1101/2022.06.29.498101>
- Ceballo, S., Piwkowska, Z., Bourg, J., Daret, A., and Bathellier, B. (2019). Targeted Cortical Manipulation of Auditory Perception. *Neuron* 104, 1168-1179.e5. <https://doi.org/10.1016/j.neuron.2019.09.043>.
- Dana, H., Chen, T.-W., Hu, A., Shields, B.C., Guo, C., Looger, L.L., Kim, D.S., and Svoboda, K. (2014a). Thy1-GCaMP6 Transgenic Mice for Neuronal Population Imaging In Vivo. *PLoS ONE* 9, e108697. <https://doi.org/10.1371/journal.pone.0108697>.
- Gămănuț, R., Kennedy, H., Toroczka, Z., Ercsey-Ravasz, M., Van Essen, D.C., Knoblauch, K., and Burkhalter, A. (2018). The Mouse Cortical Connectome, Characterized by an Ultra-Dense Cortical Graph, Maintains Specificity by Distinct Connectivity Profiles. *Neuron* 97, 698-715.e10. <https://doi.org/10.1016/j.neuron.2017.12.037>.
- Gehrlach, D.A., Weiland, C., Gaitanos, T.N., Cho, E., Klein, A.S., Hennrich, A.A., Conzelmann, K.-K., and Gogolla, N. (2020). A whole-brain connectivity map of mouse insular cortex. *ELife* 9, e55585. <https://doi.org/10.7554/eLife.55585>.
- Hennestad, E., Witoelar, A., Chambers, A.R., and Vervaeke, K. (2021). Mapping vestibular and visual contributions to angular head velocity tuning in the cortex. *Cell Reports* 37, 110134. <https://doi.org/10.1016/j.celrep.2021.110134>.
- Hong, Y.K., Lacefield, C.O., Rodgers, C.C., and Bruno, R.M. (2018). Sensation, movement and learning in the absence of barrel cortex. *Nature* 561, 542–546. <https://doi.org/10.1038/s41586-018-0527-y>.
- Liu, J., Whiteway, M.R., Sheikhattar, A., Butts, D.A., Babadi, B., and Kanold, P.O. (2019). Parallel Processing of Sound Dynamics across Mouse Auditory Cortex via Spatially Patterned Thalamic Inputs and Distinct Areal Intracortical Circuits. *Cell Reports* 27, 872-885.e7. <https://doi.org/10.1016/j.celrep.2019.03.069>.
- Makino, H., Ren, C., Liu, H., Kim, A.N., Kondapaneni, N., Liu, X., Kuzum, D., and Komiyama, T. (2017). Transformation of Cortex-wide Emergent Properties during Motor Learning. *Neuron* 94, 880-890.e8. <https://doi.org/10.1016/j.neuron.2017.04.015>.
- Milenkovic, N., Zhao, W.-J., Walcher, J., Albert, T., Siemens, J., Lewin, G.R., and Poulet, J.F.A. (2014). A somatosensory circuit for cooling perception in mice. *Nat Neurosci* 17, 1560–1566. <https://doi.org/10.1038/nn.3828>.
- Romero, S., Hight, A.E., Clayton, K.K., Resnik, J., Williamson, R.S., Hancock, K.E., and Polley, D.B. (2020). Cellular and Widefield Imaging of Sound Frequency Organization in Primary and Higher Order Fields of the Mouse Auditory Cortex. *Cerebral Cortex* 30, 1603–1622. <https://doi.org/10.1093/cercor/bhz190>.
- Scott, B.B., Thiberge, S.Y., Guo, C., Tervo, D.G.R., Brody, C.D., Karpova, A.Y., and Tank, D.W. (2018). Imaging Cortical Dynamics in GCaMP Transgenic Rats with a Head-Mounted Widefield Macrocope. *Neuron* 100, 1045-1058.e5. <https://doi.org/10.1016/j.neuron.2018.09.050>.

Referee #2

How temperature stimuli are represented in the cortex has been extensively studied in many animal models including human. Microstimulation and lesion studies in both animal and human supported an important role of both S1 and pIC in encoding temperature information. However, detailed analysis at single cell level in rodent model is still missing. In this study, the authors performed in vivo two-photon calcium imaging to examine representation of temperature stimulus in both S1 and pIC. They found that both S1 and pIC show robust responses to cooling but only pIC response to warming. They further examined the kinetics of cortical temperature responding neurons and found that they are similar to response properties reported in sensory neurons and in spinal cord neurons.

While the studies provide new cellular data supporting a critical role of the pIC in encoding temperature information, there are several concerns with regard to the premise, novelty, and conclusions.

We thank Referee 2 for their comments and for bringing up several important points that needed to be clarified in our manuscript. Overall, we take a different view to Referee 2's summary of the current state of the scientific literature concerning the cortical processing of temperature and would like to clarify the novelty of our study in more detail.

Temperature sensation has been studied for more than a century but with a central focus on the periphery of the nervous system, whereas the understanding of the cortical representation and encoding of temperature at a cellular level remained surprisingly poorly understood. The lack of clarity is partly due to stimulus delivery methods and recording techniques and activity manipulations not being optimal for an understanding of the cortical cellular encoding of non-painful thermal perception, but also because many prior studies in the cortex have examined thermal pain whereas here we examine non-painful perception. To circumvent these issues, we use precise and fast thermal stimuli (ramps up to 130°C/s) allowing tight control of thermal stimulus onset and use recording techniques and activity manipulations with relevant time and spatial resolution. Our study contributes several key findings:

- (i) The areas involved in the **cortical representation of temperature** have been investigated by a number of laboratories, but with contrasting or inconclusive results leading to statements like 'The cortical structures that mediate the sensation of cold in humans have not been clearly identified' (Greenspan et al., 2008). The majority of these studies have used functional neuroimaging, while this technique has the advantage of being non-invasive, it offers poor temporal resolution making it difficult to separate sensory related responses from other kinds of cortico-cortical or cognitive activity. Here we resolve this debate and provide physiological and behavioural evidence for pIC as the central cortical site for thermal representation. Furthermore, we clearly show that the same cortical region encodes both cool and warm.
- (ii) Previous studies on cortical thermal representation were mainly done with non-cellular resolution recording techniques, like functional neuroimaging, which prevented an understanding of the **cellular encoding of temperature**. In studies using single cell recordings techniques, non-painful cool responsive cortical neurons have been described (Clemens et al., 2018; Hellon et al., 1973; Milenkovic et al., 2014; Tsuboi et al., 1993) but, critically, extremely few warm sensitive cortical neurons have been reported. Overall, the cellular encoding of warm and cool was unclear including whether this involves two anatomically and functionally separate cellular populations following labelled lines like principles or are encoded in a more mixed temporal pattern of activity. Our data support the existence of labelled line like encoding features in the cortex.

- (iii) Often the prior focus in cortical studies has been to use non-cellular resolution techniques like EEG, fMRI or lesions that identified different regions that respond to temperature but there was no possibility to examine **cellular dynamics** with fast temporal resolution. We find a profound difference in the dynamics between cool and warm responses and an interesting difference in the cool response dynamics with some neurons reacting rapidly and others more slowly.
- (iv) While a handful of papers report cortical neurons responding to temperature, these papers mainly report whether neurons responding or not to simple thermal stimuli. Here we provide a **detailed investigation into the encoding of thermal features** like amplitude, sign (warm vs. cool) and dynamics at a cellular level and show a rich and detailed representation of temperature resembling other more established sensory modalities.
- (v) For the first time, we use fast and reversible cortical inactivation to **directly link thermal perception to cortical activity**.

Overall, we conclude that our study is both novel and fundamental to the understanding of cortical thermal encoding and thermal perception.

Major points:

1. The authors indicate in the abstract that the somatosensory cortex does not represent warming. While the difference between S1 and insular cortex is big, it is clear that the S1 also responds to warming (Extended Data Fig. 3c). Moreover, this claim is not consistent with a recent study using similar two-photon in vivo calcium imaging method, but reported robust responses to warm stimulus in S1 of awake mice (Chen XJ. et al., 2021 Journal of Neuroscience). In that study, heat stimuli was delivered using a infrared laser emitter onto the rostral back skin. The laser intensity was calibrated to heating up the skin to 40 celsius degree, which is 2 degree lower than the current study.

Thank you for bringing up this point and the study from Chen et al. (Chen et al., 2021). Below we clarify our conclusions about the representation of warm in S1 and discuss the Chen et al., (2021) study in more detail.

“it is clear that the S1 also responds to warming (Extended Data Fig. 3c)”.

While Referee 2 is correct that we did observe a few S1 neurons responding during warming stimuli, and certainly S1 cells respond to cooling, we did not mean to give the impression that S1 encodes warming in any way similar to pIC, which our data suggest is the central site for thermal encoding in the cortex. We conclude this because of different reasons:

- (i) The proportion of warm responsive neurons are strikingly different in S1 compared to pIC. In S1 only 25 from 411 (6%) responsive neurons respond to warming stimuli, in contrast, we report that 401 from 746 (54%) responsive pIC neurons responded to warming (Fig. 1 and Extended Data Fig. 3).
- (ii) The onset of the warm responses recorded in S1 vs. pIC is very different. We show in Fig. 3c, that the few warm responsive S1 neurons are significantly delayed (by around 250 ms) with respect to the shorter latency responses in pIC. In contrast, cool responses in S1 and pIC have the exact same onset time, excluding a possible methodological explanation to the delayed warm responses in S1 (Fig. 1 and Extended Data Fig. 3).
- (iii) Our data shows weak responses of a few cells during warm stimuli, but given the projection from pIC to S1 (Gămănuț et al., 2018; Gehrlach et al., 2020, Bokinec et al. 2022), we think this is an expected result and does not mean it is the primary cite for warm encoding. Indeed,

recent data has shown that primary cortical regions can encode multiple sensory modalities as a result of projections from other primary areas (e.g. Godenzini et al., 2021). We imagine that in the future, we might record some warm (or cool) responsive neurons in other cortical regions directly connected to pIC.

“Moreover, this claim is not consistent with a recent study using similar two-photon in vivo calcium imaging method, but reported robust responses to warm stimulus in S1 of awake mice (Chen XJ. et al., 2021 Journal of Neuroscience).”

Chen et al., (2021) showed responses of trunk S1 neurons during laser stimulation of the trunk skin, however there are major experimental and technical differences to our study that make it difficult to compare to our work. One experimental difference is the study of a different region in S1, the trunk representation compared to forepaw, that could be innervated in different ways, while another is the use of chlorprothixene. However, we suggest that the major difficulty in interpreting data from this study is the infrared laser emitter used for thermal stimulation. The main problem when using a laser for thermal heating is to determine the actual temperature of the skin. Chen et al. (2021) estimate skin heating due to the laser with a thermosensor placed under the skin (i.e. below the interstitial tissue, see Referee Fig. 5). This assumes the temperature of the top layers of the skin is the same as under the skin. This is very unlikely given that the skin is an effective insulator with a fat layer and, in mice, also a cutaneous muscle layer. Given that the subcutaneous temperature reached $\sim 40^{\circ}\text{C}$ during laser stimulation, and was not stable even after 5 seconds of stimulation, it is likely that the free nerve endings of C-fibers in the Dermal layer could be exposed to temperatures over the pain threshold ($>43^{\circ}\text{C}$).

Redacted

There are several other difficulties related to calibration of lasers. The heat transduction of the radiation depends on a set of physical properties including how the radiation is reflected, transmitted and absorbed, heat capacity, thermal conductivity, but these properties will vary over space in a heterogeneous structure like skin. In agreement, Plaghki and Mouraux (2003) note about the use short wavelength lasers ($<3\ \mu\text{m}$) that *'The penetrance and absorbance is not easily controlled and it is thus quite impossible to predict to which dermo-epidermal structures the energy will be confined'* (Plaghki and Mouraux, 2003). At small spatial scale and short time scale relevant for cellular encoding the radiation could therefore lead to inhomogeneous warming of the

skin. This is an important consideration given that the free nerve endings of warm sensitive C-fibers are located in the dermal layer that has a thickness of only a few microns. Taken together, we conclude that it is difficult to compare our data with Chen et al. (2021) which used a different thermal stimulator that could have delivered painful heat stimuli.

Our findings in S1 of a general lack of warm activity are further supported by two other recent papers showing no warm responding neurons in S1. In the first report, single cell electrophysiology in rat S1 oral cortical region showed a response of cortical neurons to cold water in the oral cavity in comparison to water at room temperature water, however warm water only triggered responses similar to room temperature water (Clemens et al., 2018). In another study, responding neurons in S1 were observed only when Petlier element based stimulation temperature passed the painful threshold (Osaki et al., 2022).

2. The result that the insular cortex uses separate maps for touch and temperature while the S1 has a bigger overlap is interesting but preliminary. The maps are not drawn systematically (3 body areas in the insular cortex and 1 area in the S1, Extended Data Fig. 2). The authors should at least add 2 areas to the S1 data to see if the representation of forepaw cooling is closer to touching the same body area or cooling of a different area.

Thank you for raising this point we agree that the comparative somatotopy in pIC and S1 is an interesting result. To address this more comprehensively, we used the same experimental settings to those used for pIC (Fig. 1h-j and see Methods) but implanted a cranial window centered over forepaw S1. As for pIC (Fig. 1), during S1 imaging we stimulated the hindpaw and forepaw and facial region with thermal stimuli (warm/cool) and the hindpaw and forepaw with vibrotactile stimuli. Facial tactile stimulation was avoided for both S1 and pIC imaging because it required a more specific and gentle approach to obtain clear response, most likely due to high sensitivity of facial area and movements of the face in an awake animal.

We found that:

- (i) Vibrotactile stimulation of the forepaw and hindpaw elicited spatially separated cortical responses in line with known somatotopic maps of S1 (Gămănuț et al., 2018).
- (ii) Cool responses for forepaw, hindpaw and face overlap with their respective tactile response area or with known maps (confirming Milenkovic et al. 2014).
- (iii) For all 3 body parts tested we did not observe reliable warm responses in S1.

The data are shown below in Referee Fig. 6 and as Extended Data Fig. 2.

Referee Figure 6. Somatotopic representation of temperature in forepaw S1. **b**, from left to right, cartoon schematic showing imaging setup; somatotopic map of responses locations to thermal (10°C cool) and tactile stimuli. Coloured area indicates the peak population response averaged across mice (see Methods, $n = 12$ thermal forepaw, 5 thermal hindpaw, 4 thermal face, 12 touch forepaw, 5 touch hindpaw, $n =$ mice). Data from individual mice are aligned to peak activity of the thermal forepaw response. **c**, Widefield responses to thermal stimulation of different body parts from the same dataset as in **b**. Grey lines show mean responses from individual mice (n values same as **b**), coloured lines show population mean, and grey area indicates time from start of thermal stimulus to end of plateau phase. **d**, Grey filled circles show response latencies from individual mice, coloured filled circles show mean \pm s.e.m.

The larger amplitude forepaw S1 responses to cool compared to the hindpaw and face is interesting and could reflect differences in afferent drive but an understanding of the mechanism requires further experiments. Although in the hindpaw and facial region we did not observe a response to warming in S1, small amplitude, variable signals were observed in some forepaw datasets. These weak and unreliable responses agree well with our two-photon imaging data that shows only a small minority of neurons with a variable and long latency response during warm stimulation. Because of their variability and low amplitude, they could result from a number of different non-sensory sources. Another possibility is that they result from presynaptic axonal activity from warm responsive cells in pIC that project their axons to S1. Future work could address these possibilities.

Overall, the S1 representation is primarily driven by cooling. Mice use their forepaws for haptic sensing and objects in the natural environment are typically cooler than skin temperature. We hypothesize that the temperature representation in S1 is required for the fast detection and formation of a coherent somatosensory percept during haptic sensing.

We have now changed the wording in the Results section:

Manuscript L79-80: 'Intriguingly, while responses to cooling and touch spatially overlap in S1 (Extended Data Fig. 2)⁷, they appear more separate in pIC (Fig. 1h).'

Moreover, Extended Data Fig. 3d shows only 14% of S1 neurons respond to both temperature and tactile stimuli. This number is much lower than reported by Chen XJ. et al., 2021 Journal of Neuroscience, which shows 43% of tactile responding neurons also response to heating.

While we agree these number differ across our study compared to Chen et al. (2021), it is difficult to make direct comparisons across studies which use different experimental methods, investigate different body parts and performed different analytical methods to assess significance of responses. It is known, for example, that temperature sensitivity varies widely across the body (C. Stevens Kenneth K. Choo, 1998). However, we think the crucial difference between our results and Chen et al. (2021), is probably the method of stimulation. As discussed above, Chen et al. (2021) uses laser stimulation, whereas we use a Peltier element that provided rapid and well calibrated thermal stimuli. Moreover, as discussed above, the laser stimulus used for thermal stimuli could have evoked painful thermal responses again making a comparison to our study which used a Peltier element challenging. For tactile stimulation, Chen XJ. et al., (2021) used an air puff stimulus from a liquid nitrogen tank of unknown temperature, this could mean the air stimulus itself triggered a thermal stimulus and evoked more cells than a tactile stimulus alone. In our study, we used non-thermal, mechanical actuators to obtain tactile responses. The frequency and stimulus strength are different across studies. But, because the fraction of tactile responding neurons depends on the size/amplitude of the applied force, to make a comparison across studies new experiments could be performed with identical stimulus conditions across body regions. We now cite Chen et al., (2021) in the legend of the Extended Data Fig. 3.

3. Data in Fig. 5 (optogenetics) demonstrated critical role of the pIC in temperature guided behavior. But the authors only did insula but not S1. Previous publication from the same lab have already shown pharmacological inactivation of forepaw S1 can prevent cooling perception (Milenkovic et al., 2014 Nature Neuroscience). I am curious if optogenetic silencing the S1 would also impact the behavior. if so, I think S1 should not just be a redundant circuit as claimed in current manuscript.

Thank you for bringing up this important experiment. We were sorry to hear that the referee came to the conclusion that we implied S1 was a redundant circuit in thermal processing, we certainly did not mean to give this impression. Indeed, both in this study, and in our prior work (Milenkovic et al., 2014), we show that there is a strong cool representation in S1.

To address the Referee's question, and assess the role of S1 in thermal perception further, we performed new optogenetic manipulations of S1 and pIC in VGAT-ChR2 mice trained to detect warm or cool of different amplitude (Referee Fig. 7, Extended Data Fig. 7).

Referee Figure 7. Optogenetic inhibition of S1 and pIC alters thermal perception **f**, Top, proportion of trials with licks in VGAT-ChR2 mice ($n = 4$) during optical stimulation (cyan) of S1 trials vs. without optical stimulation (blue) for different cool stimulus amplitudes. Grey lines show individual mice, coloured filled circles show mean \pm s.e.m. Middle, same as top row but for simultaneous photoinhibition of both S1 and pIC. Bottom, shows the effect of light stimulus (change in percentage of trials with licks, mean \pm s.e.m.) from mice with photoinhibition of S1 alone (yellow) or from mice with photoinhibition of both S1 and pIC simultaneously. **g**, Same as **f**, but for warm stimulation ($n = 6$ mice).

We optogenetically inhibited S1 using the same optic fiber and light stimulation parameters as for our original pIC experiments and observed a weak inhibitory impact of S1 inhibition on cool and warm detection for low amplitudes (0.5°C cooling, 2°C warming). A decreased perception of cool detection during S1 inhibition was expected, however the impact appears smaller than in our prior study (Milenkovic et al., 2014). We think this might be because previously we used pharmacological inhibition of S1 that likely had a more significant and longer lasting inhibitory effect on S1 compared to transient optogenetic inhibition presented only during the stimulus period.

The impact of S1 inhibition on cool and warm perception is also weaker compared to the same experiment in pIC (Fig. 5). We think this is because pIC is the primary thermal cortex and has a more powerful role on thermal perception than S1. To keep our experimental protocols as similar as possible we used the same diameter fiber optic for optogenetic stimulation of S1 and pIC. S1 is a larger cortical area than pIC and could contain a different local architecture, ratio of cell types and projection targets. Overall, a more detailed, mechanistic understanding of the comparative impact of optogenetic inhibition between S1 vs pIC will require detailed experiments in future studies.

Given the extremely small numbers of cells with excitatory responses to warm in S1, the inhibitory impact of S1 inhibition on low amplitude warm perception seemed counterintuitive at first thought. However, S1 optogenetic inhibition could induce many different sensory effects which have a secondary impact on perception. For example, it might alter the perception of the hand size or shape. Moreover, pIC and S1 are reciprocally connected (Gămănuț et al., 2018; Gehrlach et al., 2020, Bokinić et al. bioRxiv 2022), therefore inhibiting one area could influence sensory encoding in the other. Again, it difficult to interpret these results further without more detailed experiments addressing different and specific hypotheses.

We also went on to perform simultaneous optogenetic inhibition of S1 and pIC in trained mice using two fiber optics, one positioned directly over pIC and the other over S1. As expected, this had a far more pronounced effect on warm and cool perception than inhibiting S1 alone. Notably, however, even during dual inhibition, mice were still able to detect 10°C cooling. Our electrophysiological results above (Referee Fig. 3) show that the majority of recorded neurons are inhibited by the optogenetic inhibition, however optogenetic inhibition does not completely inhibit all spikes evoked by thermal stimuli and one possibility is that the rapid and short latency spikes evoked by cooling make the cool percept robust and resistant to manipulations and drive downstream circuits involved in perception. Previous studies have shown that subcortical circuits can also play a central role in the detection of salient, simple stimuli (Ceballo et al., 2019; Hong et al., 2018). We have observed that cool is particularly salient for mice, they are able to learn to report cooling even in the first training session and the sensory threshold is low (Paricio-Montesinos et al., 2020), therefore another possibility is that subcortical circuits, unaffected by cortical VGAT inhibition, play a major role in the detection of the more salient cool stimulus compared to warm. These hypotheses could be tested with future studies.

These data are presented in the main text.

Manuscript L281-292: 'We next tested whether the differential impact of optogenetic inhibition to warming and cooling perception resulted from the additional representation of cool in S1. We found that the impact of optogenetic inhibition of S1 is weaker compared to pIC (Extended Data Fig. 7, Fig. 5e and f), supporting the hypothesis that pIC is the central representation of temperature and could be explained by differences in cortical area size, local cellular architecture as well as projection targets. As expected, simultaneous optogenetic inhibition of S1 and pIC using fiber optics positioned directly over pIC and S1 had a more pronounced effect on warm and cool perception than inhibiting S1 alone (Extended Data Fig. 7), but was similar to the impact of inhibiting pIC alone (Fig. 5e and f). Notably, even during dual inhibition, mice were still able to detect 10°C cooling. This could result from the fast onset of cool responses providing a more robust encoding strategy or perhaps reflect the recruitment of different upstream circuits. Future experiments should be designed to address these different hypotheses.'

The relevance of the new data is mentioned in the Discussion:

Manuscript L345-347: 'One hypothesis is that the cool-only representation in S1 integrates cool with touch during haptic sensing, whereas pIC encodes the thermal sign (warm vs. cool) and level forming an independent and complete representation of thermosensation.'

Minor points:

1. The author reply quite a bit on example neurons. Figures like Extended Data Fig. 4 only have 1 neuron each. It would be better if they show all recorded neurons.

We completely agree that it is helpful to show as much raw data as possible within any study, but at some point, it becomes unmanageable to show more than a few example cells. To address this general problem, we tried to balance the presentation of raw data with population distributions in all figures so that the reader can observe both the full dataset and the example cells with graphical indications of where the cells are positioned in the relevant population distribution. For example, in Fig. 3g where we show the distribution of the 'adaptation index' for transient and sustained neurons and we show two example cool neurons illustrating a low and a high adaptation index as indicated with arrows in the histogram hopefully giving the reader an intuitive understanding of the adaptation index.

The Referee specifically mentioned Extended Data Fig. 4 which is showing the same example neurons as in Fig. 3g but with all 5 slopes that could not be shown in the main figure due to space restrictions. To improve the presentation of Extended Data Fig. 4, we now include the population averages for the temporal dynamics analysis alongside the example neurons in Extended Data Fig. 4 as it would be difficult to show all recorded neurons in the figure ($n > 400$ cells) (Referee Fig. 8). The effects in the population averages agree well with the example cell activity.

Referee Figure 8. Dynamics of thermal responses to thermal stimuli with different stimulus onset speeds. a. Example traces of transient and sustained cool (T and S) and sustained warm (S) neurons responding to stimuli with different onset speeds of approximately 130, 10, 3, 2, 1°C/s. Examples neurons are the same as presented in main Fig. 2g. Dashed lines represent averaged traces of the entire neuronal population for the transient and sustained cool (T and S) and sustained warm (S) neurons responding to stimuli with different onset speeds.

2. It is unclear if insular imaging and S1 imaging are performed in the same mice during the same session. If not, that would severely affect the conclusion as I expect different paw tethering during the different imaging sessions would result in a variability of ~1 degree.

Thank you for raising this point. Some of the data in the paper is performed in the same session some is not. The only instance in which we performed simultaneous imaging of S1 and pIC is for

the cortical pharmacological inactivation (Fig. 1f, g). In this case, we used a ‘clear skull’ preparation and imaged pIC and S1 at the same time by slightly rotating the mouse to be able to focus on both regions simultaneously. For all the other imaging (widefield and two-photon) pIC and S1 were always imaged in different sessions and animals. We have clarified this in the Methods section of the Supplementary Information:

Supplementary Information L103-109: ‘Because of physical constraints, pIC and S1 widefield imaging was performed in different animals and sessions, except during the cortical pharmacological inactivation experiments (Fig 1f, g). In this case, we used a ‘clear skull’ preparation and imaged pIC and S1 simultaneously. The setup was always adjusted to allow the paw to be placed in a similar position to all other experiments.’

Supplementary Information L144-145: ‘Because of physical constraints, pIC and S1 two-photon imaging was performed in different animals and sessions.’

The forepaw S1 region is located on the dorsal surface of the cortex, while pIC is located in a lateral region of the cortex. Because of these physical constraints, cellular resolution imaging and widefield imaging with cranial windows had to be performed in different mice. This is because our two-photon microscope only has one objective and for widefield imaging through cranial windows we ran into light orthogonal to the glass window can reflect and cause problems for imaging. Paw tethering onto the Peltier element was done identically for every mouse irrespective of the position of the cranial window or the angle at which the imaging was done. We took great care to position the Peltier element according to the rotation of the body to allow the paw a natural position on the Peltier element.

All sense organs move and, as in all studies examining sensory perception in awake mice, there will be minor differences in the exact position of the stimulus with respect to the sense organ across trials. It will introduce variability in the data and is one of the important reasons for why trials with different stimulus parameters (e.g. modality or amplitudes) are randomized. Just as one example, our data clearly show similar and reproducible cell responses are obtained by using for example 8 or 10°C cooling and warming (Fig. 4). In the light of all the potential sources of variability, we believe that the consistency of our findings means the results are more profound and conclusions strengthened and argue that possible small differences in paw placement angle will not alter our conclusions.

To address the Referee’s point, we have better described the paw positioning in the Methods section of Supplementary Information.

Supplementary Information L73-74: ‘During paw stimulation, we took care to place the fore- or hindpaw pad glabrous skin into direct contact with the center of the Peltier element.’

Supplementary Information L106-107: ‘The setup was always adjusted to allow the paw to be placed in a similar position to all other experiments.’

References

Bokiniec, P, Whitmire C. J., Leva T. M. , Poulet J. F.A. (2022) Brain-wide connectivity map of mouse thermosensory cortices. bioRxiv. doi: <https://doi.org/10.1101/2022.06.29.498101>

C. Stevens Kenneth K. Choo, J. (1998). Temperature sensitivity of the body surface over the life span. *Somatosensory & Motor Research* 15, 13–28. <https://doi.org/10.1080/08990229870925>.

- Chen, X.-J., Liu, Y.-H., Xu, N.-L., and Sun, Y.-G. (2021). Multiplexed Representation of Itch and Mechanical and Thermal Sensation in the Primary Somatosensory Cortex. *J. Neurosci.* 41, 10330–10340. <https://doi.org/10.1523/JNEUROSCI.1445-21.2021>.
- Clemens, A.M., Fernandez Delgado, Y., Mehlman, M.L., Mishra, P., and Brecht, M. (2018). Multisensory and Motor Representations in Rat Oral Somatosensory Cortex. *Sci Rep* 8, 13556. <https://doi.org/10.1038/s41598-018-31710-0>.
- Gămănuț, R., Kennedy, H., Toroczkai, Z., Ercsey-Ravasz, M., Van Essen, D.C., Knoblauch, K., and Burkhalter, A. (2018). The Mouse Cortical Connectome, Characterized by an Ultra-Dense Cortical Graph, Maintains Specificity by Distinct Connectivity Profiles. *Neuron* 97, 698-715.e10. <https://doi.org/10.1016/j.neuron.2017.12.037>.
- Gehrlach, D.A., Weiland, C., Gaitanos, T.N., Cho, E., Klein, A.S., Hennrich, A.A., Conzelmann, K.-K., and Gogolla, N. (2020). A whole-brain connectivity map of mouse insular cortex. *ELife* 9, e55585. <https://doi.org/10.7554/eLife.55585>.
- Godenzini, L., Alwis, D., Guzulaitis, R., Honnuraiah, S., Stuart, G.J., and Palmer, L.M. (2021). Auditory input enhances somatosensory encoding and tactile goal-directed behavior. *Nat Commun* 12, 4509. <https://doi.org/10.1038/s41467-021-24754-w>.
- Greenspan, J.D., Ohara, S., Franaszczuk, P., Veldhuijzen, D.S., and Lenz, F.A. (2008). Cold Stimuli Evoke Potentials That Can Be Recorded Directly From Parasyllian Cortex in Humans. *Journal of Neurophysiology* 100, 2282–2286. <https://doi.org/10.1152/jn.90564.2008>.
- Hellon, R.F., Misra, N.K., and Provins, K.A. (1973). Neurones in the somatosensory cortex of the rat responding to scrotal skin temperature changes. *The Journal of Physiology* 232, 401–411. <https://doi.org/10.1113/jphysiol.1973.sp010277>.
- Milenkovic, N., Zhao, W.-J., Walcher, J., Albert, T., Siemens, J., Lewin, G.R., and Poulet, J.F.A. (2014). A somatosensory circuit for cooling perception in mice. *Nat Neurosci* 17, 1560–1566. <https://doi.org/10.1038/nn.3828>.
- Naldaiz-Gastesi, N., Bahri, O.A., López de Munain, A., McCullagh, K.J.A., and Izeta, A. (2018). The panniculus carnosus muscle: an evolutionary enigma at the intersection of distinct research fields. *Journal of Anatomy* 233, 275–288. <https://doi.org/10.1111/joa.12840>.
- Osaki, H., Kanaya, M., Ueta, Y., and Miyata, M. (2022). Distinct nociception processing in the dysgranular and barrel regions of the mouse somatosensory cortex. *Nat Commun* 13, 3622. <https://doi.org/10.1038/s41467-022-31272-w>.
- Plaghki, L., and Mouraux, A. (2003). How do we selectively activate skin nociceptors with a high power infrared laser? Physiology and biophysics of laser stimulation. *Neurophysiologie Clinique/Clinical Neurophysiology* 33, 269–277. <https://doi.org/10.1016/j.neucli.2003.10.003>.
- Tsuboi, Y., Iwata, K., Muramatsu, H., Yagi, J., Inomata, Y., and Sumino, R. (1993). Response properties of primary somatosensory cortical neurons responsive to cold stimulation of the facial skin and oral mucous membrane. *Brain Research* 613, 193–202. [https://doi.org/10.1016/0006-8993\(93\)90899-X](https://doi.org/10.1016/0006-8993(93)90899-X).

Referee #3

This study investigates how temperature is encoded in mammalian cortex. The authors show using widefield calcium imaging that posterior insula (pIC) responds to both warming and cooling, whereas S1 responds only to cooling. Two-photon imaging shows that most pIC neurons are biased toward either warm or cold, and that the responses properties of these neurons resemble primary sensory afferents (e.g. warm responses are more sustained and encode absolute temperature, whereas cold responses rapidly adapt and encode change). The functional significance of these findings is demonstrated by optogenetically silencing pIC and showing that mice have impaired ability to report warming and cooling of the paw.

This is an interesting and timely paper that reports fundamental new information about how temperature perception is encoded in insular cortex. The experiments appear logically designed and carefully executed and the paper is clearly written. The finding that distinct pIC neurons encode warm and cold with response properties that match labelled line models from peripheral neurons is exciting.

My only major criticism is that the authors do not perform any experiments that directly test how manipulation of primary afferents or their thermosensitive channels alters the encoding of temperature in pIC. The authors' recent work (Paricio-Montesinos et al, Neuron, 2020) argues that temperature encoding in the periphery is not a simple labelled line defined by thermoTRPs but instead that TRPM8 mediates both cold and warm responses (in part via a warm induced suppression of a tonically active TRPM8+ population). Given that there is still uncertainty about how these various channels contribute to sensing of warmth, it would significantly elevate the manuscript to perform imaging in pIC in TRPM8 KO mice (or following similar manipulations) in order to test some of these models at the level of cortical dynamics.

We thank Referee 3 for their enthusiasm and support for our study. The link between thermal channels expressed in primary sensory afferent neurons and cortical thermal processing is a fascinating topic and there are certainly findings in the literature that need to be reconciled. However, our paper already provides the first investigation into the cortical cellular encoding of temperature across 5 main figures and 8 Extended Data Figures and we would prefer to keep the focus of the manuscript on this significant topic rather than extending into the role of TRPM8 and sensory afferents in cortical encoding and perception. We think that this would necessarily require a number of further experiments and figures to confirm possible findings, it would result in a shift in the emphasis of the story to the sensory periphery and not help answer unresolved questions present in our current study. In more detail, to be able to interpret possible differences in the cortical representation of temperature in the TRPM8 KO mouse, we would have to perform different experiments including:

- (i) in vivo recordings of primary sensory afferent expressing TRPM8 to examine their thermal encoding properties in vivo. To our knowledge, only a handful of TRPM8 specific afferent recordings exist from the trigeminal ganglion (Yarmolinsky et al., 2016) and this has not been addressed in the forepaw;
- (ii) acute pharmacological KO of TRPM8 in the skin and/or a conditional KO of TRPM8 in primary sensory afferent neurons to confirm any effects we observe are not a developmental issue in the constitutive knock out mouse;
- (iii) optogenetic stimulation of TRPM8 expressing neurons during cortical imaging to determine if there is a direct input to the cortex and, if so, to which cortical functional cell types (warm vs cold);

- (iv) determine the sensation that TRPM8 neurons evoke by using optogenetic stimulation of TRPM8 afferent neurons in mice trained to detect warm or cool.

These are all fascinating experiments, but without these fundamental controls, any changes in cortical activity observed in TRPM8 knock out mice would not help conclusively understand the role of TRPM8 in cortical encoding of temperature. To acknowledge this important research angle, we have included new text in the discussion stating the need for future examination into the role of different channels of sensory afferent input in the cortical encoding and perception of temperature:

Manuscript L336-341: ‘This is interesting in light of recent data suggesting interactions between warm and cool afferent pathways³⁷ and findings from the tactile pathway that suggest substantial subcortical transformation of tactile information before reaching S1³⁸. Our findings will allow future research studies to examine the separation and integration of different channels of sensory afferent input on the cortical encoding of temperature.’

References

Paricio-Montesinos, R., Schwaller, F., Udhayachandran, A., Rau, F., Walcher, J., Evangelista, R., Vriens, J., Voets, T., Poulet, J.F.A., and Lewin, G.R. (2020). The Sensory Coding of Warm Perception. *Neuron* 106, 830-841.e3. <https://doi.org/10.1016/j.neuron.2020.02.035>.

Yarmolinsky, D.A., Peng, Y., Pogorzala, L.A., Rutlin, M., Hoon, M.A., and Zuker, C.S. (2016). Coding and Plasticity in the Mammalian Thermosensory System. *Neuron* 92, 1079–1092. <https://doi.org/10.1016/j.neuron.2016.10.021>.

Reviewer Reports on the First Revision:

Referee #1 (Remarks to the Author):

The authors have addressed all my concerns and in my opinion also the main concerns of the other two reviewers comprehensively. I think that the manuscript is timely and relevant and I therefore support acceptance of the manuscript.

Referee #2 (Remarks to the Author):

I appreciate the authors' effort to address my previous concerns with new experiments and analysis.

I am not fully convinced by the authors' argument on the difference between the current manuscript vs. Chen et al., (2021) which was coming from laser stimulation. It is completely possible that more heat inputs from trunk than limb can reach S1. The authors can rule out this possibility by experiment, not argument.

I also have questions regarding new data in Extended Data Fig. 7f and g.

In Extended Data Fig. 7f, the licking trials are very different between S1 vs. S1+pIC group before light on. I bet the difference between these two control conditions is already significant. It seems like the S1+pIC group is less well trained compared to the S1 group.

In Extended Data Fig. 7g, silencing S1 has a very robust effect on behavioral responses to heating to 34 degrees. If not involved in heat sensing, why does silencing S1 have such a dramatic effect on behavioral responses to heat at this temperature?

Referee #3 (Remarks to the Author):

The revisions have improved the paper. This is an exciting study and appropriate for publication.

Author Rebuttals to First Revision:

We thank all referees for their time, support and constructive comments.

Referee 2

I appreciate the authors' effort to address my previous concerns with new experiments and analysis.

Thank you for reviewing our paper

I am not fully convinced by the authors' argument on the difference between the current manuscript vs. Chen et al., (2021) was coming from the laser stimulation. It is completely possible that more heat inputs from trunk than limb can reach S1. The authors can rule out this possibility by experiment not argument.

Referee 2 states that the lack of responses in forepaw S1 to warming compared to the laser evoked responses seen in Chen et al. (2021) could be explained by the trunk skin pathway being more sensitive to non-painful warm than the forepaw. We argue against doing further experiments to address this idea for several reasons:

(i) We examined the warm and cool representation in multiple S1 regions including forepaw, hindpaw and face, and found far stronger response to cool than to warm in each area. It is unlikely that a pathway relevant for haptic sensation (e.g. forepaw) would show such a weak warm response, whereas a region less used (e.g. trunk) would have a much stronger representation in S1.

(ii) Using the same stimulator, same stimuli and same imaging technique with which we observed a lack of warm in S1, we observe strong warm responses in pIC. Therefore, our technical approach is able to evoke cortical warm responses.

(iii) A strong S1 response to cool but a far weaker, or complete lack of, warm response is consistent with prior literature in different animal models examining different sensory pathways (Clemens et al., 2018; Hellon et al., 1973; Milenkovic et al., 2014; Tsuboi et al., 1993; Landgren, 1957, Kreisman & Zimmerman, 1973).

(iv) Even if there were a low threshold response in S1 to warm stimulation of the trunk, this would not change the main results of our paper. For the first time, we identify the cellular encoding properties of non-painful temperature, moreover, we provide causal evidence that pIC is required for thermal perception and highlight a fundamental difference in the representation of warm and cool for the representation of body parts relevant for haptic sensation (forepaw/hindpaw/face) between S1 and pIC.

(v) Rather than the trunk skin having unexpected spots of heightened warm sensitivity, it seems more likely that the responses observed in Chen et al. (2021) are the result of the differences in the stimulation method compared to our approach. In contrast to our fast, calibrated Peltier element stimulator, Chen et al. (2021) used a laser stimulator that is difficult to calibrate and could have led to activation of nociceptive thermal receptors and subsequent activation of S1 to painful heat. Chen et al. (2021) estimate skin heating to the laser with a thermosensor placed under the skin. This assumes the temperature of the top layers of the skin is the same as the temperature under the skin. This is unlikely given that the skin is an effective insulator with a fat layer and, in mice, also a cutaneous muscle layer. Given that the subcutaneous temperature reached ~40°C during laser stimulation in Chen et al. (2021), and was not stable even after 5 seconds of stimulation, it is likely that the free nerve endings of C-fibers in the dermal layer could be exposed to temperatures over the pain threshold (>43°C).

We also mentioned that there are several other difficulties related to calibration of lasers. The heat transduction of the radiation depends on a set of physical properties including how the radiation is reflected, transmitted and absorbed, heat capacity, thermal conductivity, but these properties will vary over space in a heterogeneous structure like skin. In agreement, Plaghki and Mouraux (2003) note about the use short wavelength lasers ($<3 \mu\text{m}$) that 'The penetrance and absorbance is not easily controlled and it is thus quite impossible to predict to which der-mo-epidermal structures the energy will be confined' (Plaghki and Mouraux, 2003). At small spatial scale and short time scale relevant for cellular encoding the radiation could therefore lead to inhomogeneous warming of the skin. This is an important consideration given that the free nerve endings of warm sensitive C-fibers in the dermal layer have a thickness of only a few microns.

(vi) To perform the series of experiments required to address warm processing in trunk S1 would be a time-consuming study in itself. Stimulation of the trunk with a calibrated Peltier element in an awake mouse would be challenging and require a new trunk-fixation method. Moreover, it would require additional control experiments to confirm that hair removal on the trunk does not cause irritation of the skin and altered pain thresholds. If we then wanted to go on and directly compare our data to those from Chen et al. (2021), we would need to use the experimental protocol from this lab including the same laser, the same calibration method and same anesthesia.

Overall, we used a calibrated thermal stimulator on non-anesthetized mice and provide consistent results across different body parts and different cortical regions. Therefore, we argue that further experiments to address differences in the thermal threshold between the trunk and forepaw/hindpaw/face pathways and to compare our results more closely to those from Chen et al. (2021) are unnecessary to reach the conclusions made in our study.

I also have questions regarding new data in Extended Data Fig.7f and g.

In Extended Data Fig.7f, the licking trials is very different between S1 vs. S1+pIC group before light on. I bet the difference between these two control condition is already significant. It seems like the S1+pIC group is less well trained in compare to S1 group.

The data in main Figure 5 and Extended Data Figure 7 and 8 show that optogenetically inhibiting pIC alters thermal perception. To address 2 referees' questions about the role of S1 in thermal perception, we performed additional experiments to optogenetically inhibit S1 alone and S1+pIC together during thermal perception task. In Extended Data Fig.7f, one mouse has lower performance at different stimulus amplitudes, and there is a mouse in the S1 dataset with low performance at 31°C . Some variability in behavioral experiments is expected and does not change the overall conclusions that optogenetic inhibition of S1 has an effect on thermal detection, and that combined inhibition of S1+pIC has a larger effect than S1 alone. There are a number of reasons that support this statement:

(i) There is minimal to no effect of optogenetic inhibition for 10°C cooling in both S1+pIC and S1 datasets.

(ii) We followed the same inclusion criteria for all groups throughout our paper ($>70\%$ performance the day before testing, see Methods). Despite this, it is not possible to reduce all variability on the subsequent test day.

(iii) Visually inspecting data from individual mice shows that all mice drop in performance when manipulating S1 + pIC (for stimuli $<10^{\circ}\text{C}$ cooling), whereas the results are more mixed for S1 alone.

(iv) Taking the lowest performing mouse out of the S1+pIC dataset would increase the difference between S1+pIC vs S1 groups and would strengthen, rather than weaken, the overall conclusion that the impact of S1+pIC is stronger than S1 alone.

In Extended Data Fig.7g, silencing S1 have very robust effect on behavioral responses to heating to 34 degree. If not involve in heat sensing, why silencing S1 have such dramatic effect on behavioral responses to heat at this temperature.

Photoinhibition of S1 could impact the reporting of the small amplitude thermal stimulus in many ways unrelated to the perception of warmth. For example, the perception of paw size or surface texture could be altered leading to a change in reporting accuracy, especially at low stimulus amplitudes that are harder to detect. It is also possible that photoinhibition of S1 alters warm processing in pIC via direct or multisynaptic pathways between S1 and pIC or from S1 to higher cortical regions. Whatever the exact reason for this small effect on low amplitude warm stimuli, it does not change the central result that very few neurons respond to warm in S1 and that pIC has a profound effect on warm perception. Understanding the precise role of S1 in thermal perception, and why it contains a representation of cold and not warm, is an extremely interesting and important research topic that would require a full study in itself and cannot be address by optogenetic manipulation experiments alone.

References

- Chen, X.-J., Liu, Y.-H., Xu, N.-L., and Sun, Y.-G. (2021). Multiplexed Representation of Itch and Mechanical and Thermal Sensation in the Primary Somatosensory Cortex. *J. Neurosci* 41, 10330–10340.
- Clemens, A.M., Fernandez Delgado, Y., Mehlman, M.L., Mishra, P., and Brecht, M. (2018). Multisensory and Motor Representations in Rat Oral Somatosensory Cortex. *Sci Rep* 8, 13556.
- Hellon, R.F., Misra, N.K., and Provins, K.A. (1973). Neurones in the somatosensory cortex of the rat responding to scrotal skin temperature changes. *J Physiol* 232, 401–411.
- Kreisman, N.R. and Zimmerman, I.D. (1973). Representation of information about skin temperature in the discharge of single cortical neurons *Brain Res* 55, 343–353.
- Landgren, S. (1957). Cortical reception of cold impulses from the tongue of the cat. *Acta Physiol Scand* 40, 202–209.
- Milenkovic, N., Zhao, W.-J., Walcher, J., Albert, T., Siemens, J., Lewin, G.R., and Poulet, J.F.A. (2014). A somatosensory circuit for cooling perception in mice. *Nat Neurosci* 17, 1560–1566.
- Plaghki, L., and Mouraux, A. (2003). How do we selectively activate skin nociceptors with a high power infrared laser? *Physiology and biophysics of laser stimulation. Neurophysiologie Clinique/Clinical Neurophysiology* 33, 269–277.
- Tsuboi, Y., Iwata, K., Muramatsu, H., Yagi, J., Inomata, Y., and Sumino, R. (1993). Response properties of primary somatosensory cortical neurons responsive to cold stimulation of the facial skin and oral mucous membrane. *Brain Research* 613, 193–202.